# Assessing and visualizing fragility of clinical results with binary outcomes in R using the fragility package

Lifeng Lin[1]*, Haitao Chu[2,3]

**1** Department of Statistics, Florida State University, Tallahassee, FL, United States of America, **2** Statistical Research and Innovation, Global Biometrics and Data Management, Pfizer Inc., New York, NY, United States of America, **3** Division of Biostatistics, University of Minnesota School of Public Health, Minneapolis, MN, United States of America

* linl@stat.fsu.edu

**Data Availability Statement:** All datasets used in this article are available in the R package "fragility" (https://cran.r-project.org/package=fragility).

**Funding:** This research was supported in part by the US National Institutes of Health/National

## Abstract

With the growing concerns about research reproducibility and replicability, the assessment of scientific results' fragility (or robustness) has been of increasing interest. The fragility index was proposed to quantify the robustness of statistical significance of clinical studies with binary outcomes. It is defined as the minimal event status modifications that can alter statistical significance. It helps clinicians evaluate the reliability of the conclusions. Many factors may affect the fragility index, including the treatment groups in which event status is modified, the statistical methods used for testing for the association between treatments and outcomes, and the pre-specified significance level. In addition to assessing the fragility of individual studies, the fragility index was recently extended to both conventional pairwise meta-analyses and network meta-analyses of multiple treatment comparisons. It is not straightforward for clinicians to calculate these measures and visualize the results. We have developed an R package called "fragility" to offer user-friendly functions for such purposes. This article provides an overview of methods for assessing and visualizing the fragility of individual studies as well as pairwise and network meta-analyses, introduces the usage of the "fragility" package, and illustrates the implementations with several worked examples.

## Introduction

Research reproducibility and replicability have been major concerns in many areas of scientific research [1–6]. Such issues may be largely owing to the misuse of *p* values [7, 8], which are often misinterpreted as a measure of treatment effects in clinical studies [9, 10]. Consequently, studies with smaller *p* values (i.e., statistically more significant effects) are more likely to be published; this phenomenon is often referred to as publication and selective reporting bias or small-study effects [11–18]. This bias may distort clinical conclusions toward an artificially favorable direction and thus greatly threaten their reliability. Due to these concerns, communities across many scientific fields have recently called for more careful interpretations of *p* values and statistical significance [19–22]. In an effect to reduce publication bias, it has been

Institute of Mental Health grant R03 MH128727, National Institutes of Health/National Library of Medicine grant R01 LM012982 (LL and HC), and National Institutes of Health/National Center for Advancing Translational Sciences grant UL1 TR001427 (LL). The content is solely the responsibility of the authors and does not necessarily represent the official views of the National Institutes of Health. The funders had no role in study design, data collection and analysis, decision to publish, or preparation of the manuscript.

**Competing interests:** The authors have declared that no competing interests exist.

recommended to pre-register clinical trials or publish their protocols before obtaining the final results [23].

To supplement the use of $p$ values and 95% confidence intervals (CIs) for assessing treatment effects in clinical studies with binary outcomes, Walsh et al. [24] proposed the fragility index (FI) to quantify their fragility (or robustness). The FI is defined as the minimal event status modifications that can alter a study result's statistical significance. For example, if an originally significant treatment effect estimate becomes non-significant by modifying only a single patient's event status (e.g., from no disease to disease), then the clinical study's conclusion is highly fragile. In this case, clinicians may need to carefully borrow real-world evidence to assess the likelihood of that patient developing the disease and appraise the reliability of the evidence about treatment effects from this study. Similar concepts have also been considered in the earlier literature [25, 26]. With the growing concerns about research reproducibility and replicability, the FI has regained much attention in recent years. It has been applied to assess the fragility of randomized controlled trials in several clinical areas, such as anticancer medicine, critical care, surgery medicine, and obstetrics [27–33].

The concerns of research reproducibility and replicability also arise in systematic reviews and meta-analyses (MAs). Publications of MAs have been rapidly increasing in the past few decades, because they offer a powerful tool for synthesizing and contrasting existing findings and producing more precise effect estimates [34, 35]. However, sometimes different MAs focusing on the same topic can have inconsistent conclusions [36–38]. Similar to pre-registering clinical trials, pre-registered prospective MAs have been recommended [39, 40]. Recently, the FI was extended to assess the fragility of conventional pairwise MAs as well as network meta-analyses (NMAs) of multiple treatment comparisons [41, 42]. The FI of an MA is defined similarly to that in a clinical trial; however, its estimation is more complicated, because the modifications of event status may occur in different studies within the MA. Therefore, it is computationally challenging for applied scientists to calculate and interpret the FI of an MA.

It may not be sufficient to rely completely on the numerical value of the FI derived at a specific significance level (e.g., commonly used 0.05) for properly interpreting the fragility. For example, the FI may be highly associated with the $p$ value under certain settings [43]. In such cases, the FI may not provide much more information in addition to the $p$ value. Nevertheless, the correlation between the FI and $p$ value is generally expected, because the FI is derived based on $p$ value (or CI); however, as long as the correlation coefficient is not nearly ±1, the FI can still serve as a useful supplement. Its interpretation of "the number of events modified for altering significance" is intuitive for clinicians. This is similar to the common practice of reporting point estimates of treatment effects, their standard errors (SEs), $p$ values, and CIs; each of them provides important information for assessing treatment comparisons, although they are associated with one another. Moreover, no widely-accepted guidelines are available to evaluate the extent of fragility based on the FI value [44]. Clinicians' opinions (e.g., about the clinical importance of an effect) may be incorporated when assessing the fragility [45, 46]. For example, it is likely that a non-event may be changed to be an event for common diseases, but it is less likely for rare diseases. In addition, the FI may not be very suitable for analyses of time-to-event data, in which the timing of events, rather than the occurrence of events, is of primary interest [47–49]. In summary, as a relatively new measure, more comprehensive evaluations, including visualizations of the whole process that alters the significance, should be taken into account when interpreting the FI in clinical practice.

To the best of our knowledge, very limited software packages are available for assessing the fragility of clinical results, and no package has been developed yet for visualizing the fragility. An online calculator (https://clincalc.com/Stats/FragilityIndex.aspx) offers a simple tool to calculate the FI of individual studies; users only need to input the event counts and sample sizes

in the two treatment groups in a clinical study. However, it does not provide options for specifying the statistical significance level, statistical method used for deriving the significance, etc. The significance level is fixed at 0.05, and Fisher's exact test is the only option to derive the FI. An R package "fragilityindex" [50] is also available to calculate the FI of individual studies; it additionally extends the FI to logistic regression analyses and survival data analyses. Nevertheless, it only permits users to specify the significance level; many other important factors (such as treatment groups in which event status is modified) that may impact the FI cannot be changed. Atal et al. [41] provide a web interface to calculate the FI of a pairwise MA (https://clinicalepidemio.fr/fragility_ma/); the Stata module "metafrag" [51] can also be used for this purpose.

We have developed an R package "fragility" [52] that provides many additional options for assessing and visualizing the fragility of individual trials, pairwise MAs, and NMAs. This article gives an overview of these options and introduces the usage of the "fragility" package in detail with several worked examples. The remaining content is organized as follows. First, we review methods for assessing the fragility in various clinical settings. Second, we introduce the structures of different types of datasets and the usage of various functions provided by the "fragility" package. Third, we present several worked examples and display their results to illustrate the usage of these functions. Finally, we provide a brief discussion about future improvements.

## Materials and methods

### Assessing and visualizing the fragility

**Fragility of an individual clinical study.**   Suppose that a clinical study compares two treatments, denoted by 0 and 1, with a binary outcome. The results are typically reported in a 2×2 table (Table 1). Let $n_0$ and $n_1$ be the sample sizes in treatment groups 0 and 1, respectively, and $e_0$ and $e_1$ be the event counts. These counts are non-negative integers, and $e_0 \leq n_0$ and $e_1 \leq n_1$.

By modifying some events' status, the impact on the study result can be used to calculate the FI. The uncertainties in event status are common in practice; for example, if the follow-up periods for some participants are not sufficient, their disease outcomes may occur after the end of study. [24] originally proposed to assess the fragility of a study by modifying event status only in a single treatment group; such a group is chosen as the one with the fewest events. Nevertheless, this restriction may not guarantee that the modifications of event status for altering statistical significance or non-significance are minimal. In general, we may consider event status modifications in both treatment groups, as in Table 1. Specifically, let $f_0$ and $f_1$ be the numbers of non-events changed to events in groups 0 and 1, respectively. They may take any integer values between $-e_k$ and $n_k - e_k$ ($k = 0, 1$). Negative values of $f_0$ or $f_1$ indicate decreasing

**Table 1. Illustration of a 2×2 table and event status modifications.**

| Treatment | Event | Non-event | Sample size |
|---|---|---|---|
| 2×2 table of the original study: | | | |
| Group 0 | $e_0$ | $n_0 - e_0$ | $n_0$ |
| Group 1 | $e_1$ | $n_1 - e_1$ | $n_1$ |
| 2×2 table with event status modifications: | | | |
| Group 0 | $e_0 + f_0$ | $n_0 - e_0 - f_0$ | $n_0$ |
| Group 1 | $e_1 + f_1$ | $n_1 - e_1 - f_1$ | $n_1$ |

event counts in the corresponding group, while positive values indicate increasing event counts; setting $f_0$ or $f_1$ to 0 implies no event status modification.

Many statistical methods can be used to assess the association between a treatment and an outcome in a 2 × 2 table [53]. Fisher's exact test is commonly used for this purpose; its $p$ value is calculated based on a hypergeometric distribution under the null hypothesis. This test is particularly useful for small sample sizes, because many alternative methods use large-sample asymptotic properties and may not perform well for small sample sizes. The chi-squared test is another popular method, and its $p$ value is based on the asymptotic chi-squared distribution under the null hypothesis; thus, this test generally requires sufficiently large sample sizes.

Clinicians also frequently use certain measures to quantify treatment effects for binary outcomes, e.g., the odds ratio (OR), relative risk (RR), and risk difference (RD); $p$ values may be produced based on these effect sizes. Without loss of generality, these effect sizes are calculated for the comparison of group 1 vs. group 0 throughout this article. The OR and RR are conventionally analyzed on a logarithmic scale for a better approximation to the normal distribution. Specifically, the log OR is estimated as

$$y(f_0, f_1) = \log \frac{(e_1 + f_1)/(n_1 - e_1 - f_1)}{(e_0 + f_0)/(n_0 - e_0 - f_0)}$$

with SE

$$s(f_0, f_1) = \left( \frac{1}{e_0 + f_0} + \frac{1}{n_0 - e_0 - f_0} + \frac{1}{e_1 + f_1} + \frac{1}{n_1 - e_1 - f_1} \right)^{1/2}.$$

The log RR is estimated as

$$y(f_0, f_1) = \log \frac{(e_1 + f_1)/n_1}{(e_0 + f_0)/n_0}$$

with SE

$$s(f_0, f_1) = \left( \frac{1}{e_0 + f_0} + \frac{1}{e_1 + f_1} - \frac{1}{n_0} - \frac{1}{n_1} \right)^{1/2}.$$

The RD is estimated as

$$y(f_0, f_1) = \frac{e_1 + f_1}{n_1} - \frac{e_0 + f_0}{n_0}$$

with SE

$$s(f_0, f_1) = \left[ \frac{(e_0 + f_0)(n_0 - e_0 - f_0)}{n_0^3} + \frac{(e_1 + f_1)(n_1 - e_1 - f_1)}{n_1^3} \right]^{1/2}.$$

In the presence of zero counts, a continuity correction (often 0.5) needs to be applied to all data cells in the 2×2 table for producing these estimates [54].

Consequently, a certain set of event status modifications $f_0$ and $f_1$ leads to a $p$ value based on each of the above five methods for assessing the association between the treatment and outcome, denoted by $p(f_0, f_1)$. The $p$ value of the original study with no event status modification is $p(0, 0)$ with $f_0 = f_1 = 0$. For the chi-squared test, OR, RR, and RD, their $p$ values may not be accurate when some data cells are small, because they all use large-sample asymptotic null

distributions to calculate $p$ values. The estimated log OR, log RR, and RD are assumed to approximately follow the normal distribution, so their $p$ values are calculated as $p(f_0, f_1) = 2\Phi\left(-\frac{|y(f_0,f_1)|}{s(f_0,f_1)}\right)$ (two-sided) or $p(f_0, f_1) = \Phi\left(-\frac{|y(f_0,f_1)|}{s(f_0,f_1)}\right)$ (one-sided), where $\Phi(\cdot)$ denotes the cumulative distribution function of the standard normal distribution. The OR, RR, and RD can indicate the direction of treatment effects; thus, the alternative hypothesis could be two- or one-sided. Fisher's exact test and the chi-squared test evaluate the association with no specific direction; therefore, their $p$ values are two-sided.

For each method, the $p$ values $p(f_0, f_1)$ based on all considered event status modifications can be visualized as a matrix of points; each point represents a $p$ value, with the x- and y-axes representing its corresponding event status modifications, and its color distinguishes the magnitude of the $p$ value [55]. When event status modifications are restricted to a single treatment group, the $p$ values $p(f_0, 0)$ or $p(0, f_1)$ can be presented against $f_0$ or $f_1$ in a scatterplot for visualizing the change of $p$ values as event status modifications vary. These plots will be illustrated in our worked examples later.

Assume the statistical significance level is pre-specified at $\alpha$. Formally, if the original study result is statistically significant with $p(0, 0) < \alpha$, then the FI is defined as

$$\text{FI} = \min_{p(f_0,f_1) \geq \alpha} \{|f_0| + |f_1|\};$$

if the original study result is non-significant with $p(0, 0) \geq \alpha$, then the FI is

$$\text{FI} = \min_{p(f_0,f_1) < \alpha} \{|f_0| + |f_1|\}.$$

A smaller value of FI indicates a more fragile result. The above minimization problems are subject to $-e_k \leq f_k \leq n_k - e_k$ ($k = 0, 1$). These ranges could be adjusted to accommodate with clinicians' needs. For example, if it is more likely that some events are not observed, then one may restrict the ranges to be non-negative for yielding more events. One may also restrict event status modifications to a single group, as in Walsh et al. [24]. When the modifications are restricted to group 0, the resulting FI is

$$\text{FI}_0 = \begin{cases} \min_{p(f_0,0) \geq \alpha}|f_0| & \text{if } p(0,0) < \alpha; \\ \min_{p(f_0,0) < \alpha}|f_0| & \text{if } p(0,0) \geq \alpha. \end{cases}$$

Similarly, when the modifications are restricted to group 1, the resulting FI is

$$\text{FI}_1 = \begin{cases} \min_{p(0,f_1) \geq \alpha}|f_1| & \text{if } p(0,0) < \alpha; \\ \min_{p(0,f_1) < \alpha}|f_1| & \text{if } p(0,0) \geq \alpha. \end{cases}$$

Clearly, $1 \leq \text{FI} \leq \min\{\text{FI}_0, \text{FI}_1\}$. It is possible that the significance or non-significance cannot be altered based on given ranges of event status modifications; in such cases, we define FI as not available (NA). This may happen when sample sizes are small, as they only permit a narrow range of modifications.

Of note, this article discusses multiple methods for testing the association between treatment and outcome and thus deriving the FI. We hope that they offer flexibility for users when assessing clinical studies' fragility based on different tools. We do not suggest that users should try all methods because this practice could lead to "fragility-hacking." Users are recommended to use the statistical method specified in the study protocol for deriving the FI.

Although the significance level is conventionally set at $\alpha = 0.05$, this choice is arguably arbitrary, and the resulting false positive rate may be considered high in some fields of science. Many researchers propose to lower this standard to $\alpha = 0.005$ for improving research reproducibility and replicability [56, 57]. As the FI is derived based on a specific significance level, the significance level should always be reported alongside the associated level. Instead of relying on the FI at a single significance level, clinicians might also be interested in the trend of the FI as the significance level varies (e.g., from 0.005 to 0.05), which can be visualized in a scatterplot [55]. Theoretically, the FI is a function of the significance level, denoted by FI($\alpha$). This is a step function because the FI must take positive integer values. Suppose the FI is evaluated from $\alpha = \alpha_L$ (say, 0.005) to $\alpha = \alpha_U$ (say, 0.05). We may consider the average of the area under the function to quantify the overall fragility among the range of significance levels [$\alpha_L$, $\alpha_U$]. The idea is similar to the area under the receiver operating characteristic curve (AUC) used in diagnostic decision-making. The average FI is $\text{FI}_{\text{avg}} = \frac{1}{\alpha_U - \alpha_L} \int_{\alpha_L}^{\alpha_U} \text{FI}(\alpha) d\alpha$. In practice, this quantity can be approximated by the average of FIs at $B$ (say, 100) equally-spaced values between $\alpha_L$ and $\alpha_U$, denoted by $\alpha_b$ for $b = 1, 2, \ldots, B$ with $\alpha_1 = \alpha_L$ and $\alpha_B = \alpha_U$. Because $\int_{\alpha_L}^{\alpha_U} \text{FI}(\alpha) d\alpha \approx \frac{\alpha_U - \alpha_L}{B} \sum_{b=1}^{B} \text{FI}(\alpha_b)$ for a sufficient large $B$, the average FI is $\text{FI}_{\text{avg}} \approx B^{-1} \sum_{b=1}^{B} \text{FI}(\alpha_b)$, i.e., the arithmetic mean of the values of FI($\alpha_b$).

Multiple clinical studies may be conducted on the same topic; they compare the same treatment groups and investigate the same outcome. Clinicians may want to compare the fragility across the multiple studies. As the FI of an individual study depends on the sample size, it might not be sensible to directly compare the FIs of the multiple studies. Alternatively, one may use the relative measure, fragility quotient (FQ), to compare the multiple studies' fragility [58]. Specifically, $\text{FQ} = \frac{\text{FI}}{n_0 + n_1} \times 100\%$, where $n_0 + n_1$ is the total sample size of the study. Thus, the FQ represents the minimal percentage change of event status among all participants that can alter the significance (or non-significance), and it ranges within 0%–100%.

**Fragility of a meta-analysis.** An MA aims at synthesizing and contrasting findings from multiple independent studies on the same topic. Consider an MA with a binary outcome that contains $N$ studies. Each study compares the same two treatment groups (denoted by 0 and 1) and reports its 2×2 table with event counts $e_{i0}$ and $e_{i1}$ and sample sizes $n_{i0}$ and $n_{i1}$ in the two groups ($i = 1, \ldots, N$). The effect measure can be the (log) OR, (log) RR, or RD. Let $y_i$ and $s_i$ be the estimated effect size and its SE, respectively, in study $i$. The continuity correction is applied to studies with zero data cells. The estimated effect sizes are conventionally assumed to approximately follow the normal distributions $y_i \sim N(\theta_i, s_i^2)$ within studies, where $\theta_i$ denotes the underlying true effect size of study $i$.

Here, the within-study SEs $s_i$ are assumed to be fixed, known values. Alternative exact methods (without the approximation to the normal distributions) are available via generalized linear mixed models or Bayesian hierarchical models; they can avoid the continuity correction in the presence of zero data cells and may have better performance than the conventional method for sparse data [59–63]. However, to assess the fragility of the MA, this article focuses on the conventional method instead of the alternatives, because many iterations may be needed to derive the FI, and it may be computationally demanding to repeat the exact methods many times. Also, as most MA applications have used the conventional method so far, the FI derived from this method may better reflect the current practice.

The underlying true effect sizes are further assumed to follow the normal distribution $\theta_i \sim N(\theta, \tau^2)$, where $\tau^2$ is the between-study variance owing to heterogeneity. A special case is that $\tau^2 = 0$, which implies $\theta_i = \theta$ for all studies; this case is referred to as the fixed-effect or common-effect setting, and $\theta$ represents the common effect size shared by all studies. On the other

hand, $\tau^2 > 0$ yields the random-effects setting, where $\theta$ is interpreted as the overall effect size across studies. In both settings, $\theta$ is of primary interest, and the MA aims at estimating this parameter and its CI. One may refer to Borenstein et al. [64], Riley et al. [65], and many other articles for extensive discussions about the interpretation and selection of the fixed-effect and random-effects settings.

The between-study variance $\tau^2$ plays a critical role in the random-effects MA because it greatly impacts the CI of the treatment effect estimate and thus the statistical significance. It can be estimated via several approaches. The DerSimonian–Laird (DL) estimator by [66] is the most popular one; nevertheless, several better alternatives, e.g., the restricted maximum likelihood (REML) estimator, have been shown to perform better in general [67, 68]. Let $\hat{\tau}^2$ be the estimated between-study variance; under the fixed-effect setting, set $\hat{\tau}^2 = 0$. Each study in the MA is assigned with a weight $w_i = 1/(s_i^2 + \hat{\tau}^2)$. The overall effect size is estimated as

$$\hat{\theta} = \frac{\sum_{i=1}^{N} w_i y_i}{\sum_{i=1}^{N} w_i}.$$

It approximately follows the normal distribution, and its $(1 - \alpha) \times 100\%$ CI is conventionally constructed as

$$\hat{\theta} \pm z_{1-\alpha/2} \times \left(\sum_{i=1}^{N} w_i\right)^{-1/2},$$

where $z_{1-\alpha/2}$ denotes the $1 - \alpha/2$ quantile of the standard normal distribution. Alternatively, [69, 70] refined the CI by accounting for the variation in $\hat{\tau}^2$. The Hartung–Knapp–Sidik–Jonkman (HKSJ) method constructs the CI as

$$\hat{\theta} \pm t_{N-1,1-\alpha/2} \times \left\{\frac{\sum_{i=1}^{N} w_i(y_i - \hat{\theta})^2}{(N-1)\sum_{i=1}^{N} w_i}\right\}^{1/2},$$

where $t_{N-1,1-\alpha/2}$ denotes the $1 - \alpha/2$ quantile of the $t$ distribution with $N - 1$ degrees of freedom. It has been shown to have a better coverage probability than the CI based on the normal distribution, especially when the number of studies $N$ is small [71].

To assess the fragility of an MA, an ideal approach is to exhaustively enumerate all possible event status modifications step by step; however, this procedure may be impractical from the computational perspective if many steps are needed to alter the significance or non-significance. Suppose that the overall effect size is significant and is above the null value. At each step of modifying event status, we may need to consider decreasing one event count in group 1 or increasing one event count in group 0 in a single study; thus, assuming that the event counts have not achieved the bounds (i.e., 0 or sample size), there are $2N$ possible cases for this step. Such iterations will terminate only after the significance is altered, so we need to perform up to $(2N)^{\mathrm{FI}}$ MAs during this process. This is not practical in many real-world applications; for example, even for a relatively small MA with $N = 10$ studies, if the FI is 5, then this exhaustive search needs to perform over 3 million different MAs with modified event status.

Instead of enumerating all possible event status modifications, Atal et al. [41] proposed a heuristic iterative process based on the CI of the overall effect size estimate to derive the FI. Specifically, suppose that the original MA yields a significant overall effect size estimate, and it is larger than the null value. We initiate the iterative process from the original MA (step 0). In order to move the CI toward the null value, event status is modified to decrease event counts (down to 0) in group 1 or increase those in group 0 (up to the corresponding sample size). At each step, one event is changed to a non-event in group 1, or one non-event is changed to an

event in group 0 in a certain study; separate MAs are performed based on the data with each of the above modifications to produce the CIs of the overall effect size estimate. The modification that leads to the smallest lower bound of the CI (i.e., the one closest to the null value if the CI still does not cover it) is selected as the optimal one for facilitating the process of altering the significance. Based on the optimal modifications identified in the previous steps, the iterations continue until the CI covers the null value. Because each step contains up to $2N$ modifications, the above algorithm only needs to perform up to $2N \times \text{FI}$ MAs to derive the FI, making the process computationally feasible. This number is much smaller than $(2N)^{\text{FI}}$ in the exhaustive search, especially when $N$ or the FI value is large. For visualizations of the iterative steps for deriving the FI of the MA, readers may refer to Fig 2 in Atal et al. [41]. We will also provide a worked example later to demonstrate the process.

On the other hand, suppose that the original MA has a non-significant overall effect size estimate. Unlike the case of a significant overall effect size estimate where the CI is moved toward only one specific direction, now the CI covers the null value, and we may move it toward either the left or right direction for achieving significance. For each direction, a separate FI can be derived via an algorithm similar to the one described above; the final FI is the minimum value of these two FIs.

In cases that significance or non-significance cannot be altered, the FI is defined as NA. The FQ can be similarly calculated for the MA; it is the FI divided by the total sample size across all studies. To visualize the process of the iterative algorithm for deriving the FI, one may present the changes in event counts in the two treatment groups along with the studies involved in the corresponding modifications against the iterations; we will provide worked examples to illustrate the visualizations.

**Fragility of a network meta-analysis.** NMA is an extension of the conventional pairwise MA that compares only a pair of treatments at one time; it aims at comparing multiple treatments simultaneously by synthesizing both direct and indirect evidence about treatment comparisons [72, 73]. Suppose a trial compares treatments A and C and another trial compares B and C; these two trials provide indirect evidence for A vs. B via the common comparator C. NMA has been increasingly used in recent years, because many treatments may be available for a specific disease outcome. It is particularly useful when some treatments of interest (e.g., new drugs) have been seldom compared directly, but many trials have compared them with some common treatments (e.g., placebo). It may produce more precise treatment effect estimates than separate pairwise MAs and provide a coherent treatment ranking for decision making [74–77].

Various methods have been developed to perform NMA under both the frequentist and Bayesian frameworks [78–87]. To assess the fragility of an NMA, similar iterative procedures for a pairwise MA can be used [42]. We focus on the frequentist method by Rücker [79] to produce the CIs of treatment comparisons in the NMA. Although in theory any method can be used to derive the FI, the Bayesian methods could be very time-consuming even for analyzing a single NMA, so it may not be practical to iteratively apply them to many NMAs with modified event status.

Specifically, unlike the case of a pairwise MA that involves a single treatment comparison, the NMA contains multiple comparisons, each yielding a separate effect size estimate. Let $K$ be the number of treatments in the NMA; a total of $K(K-1)/2$ comparisons are estimated. Therefore, the FI is not defined for the whole NMA as in individual studies or pairwise MAs; it is defined for each treatment comparison. Consequently, for a specific pair of treatments, say A and B, we consider event status modifications based on the significance of their comparison B vs. A. Modifying any event status, even for those not in groups A and B, may change the results of all treatment comparisons; thus, in theory, the event status modifications are possible for

each study's each treatment group. However, this would dramatically increase the computation time. Also, it is intuitive to modify event status directly in groups A and B, and such modifications are expected to have a larger impact on the estimated effect size of B vs. A and can alter the significance or non-significance faster. Therefore, during each iteration for deriving the FI for B vs. A, this article only considers event status modifications in these two groups. For example, if the effect size of B vs. A is significantly larger than the null value in the original NMA, then in each iteration, we consider decreasing event counts in group B or increasing those in group A in certain studies until the significance is altered.

Similar to assessing the fragility of an individual study and a pairwise MA, the FI of an NMA is defined as NA if the significance or non-significance cannot be altered. Of note, as mentioned above, the calculation of the NMA's FI for comparison B vs. A is based only on modifying event status in groups A and B. It is possible that the change of significance cannot be achieved by any event status modification in groups A and B, but it could be achieved by modifications in other groups. Therefore, users should interpret an FI value of NA in the contexts of the event status modifications in the relevant two groups A and B only.

The process of deriving the FI can also be visualized for each treatment comparison using a similar approach for a pairwise MA. The relative measure FQ can be calculated as the FI divided by the sample size, but it may have two versions in the NMA. It seems straightforward to use the total sample size $n_{\mathrm{NMA}}$ across all studies and all treatment groups in the whole NMA as the denominator for calculating the FQ. However, the FQ derived in this way has an upper bound $\frac{n_{\mathrm{AB}}}{n_{\mathrm{NMA}}} \times 100\%$, where $n_{\mathrm{AB}}$ denotes the sample size in groups A and B across all studies, because the algorithm only modifies event status in the associated two treatments for a specific comparison. This upper bound differs for different pairs of treatments, implying a methodological limitation. Alternatively, for the comparison B vs. A, we may calculate the FQ as the FI divided by $n_{\mathrm{AB}}$, so that this FQ still ranges within 0%–100% and could be fairly compared across treatment pairs.

## Using the R package "fragility"

The source file of the R package "fragility" and its manual are available on the Comprehensive R Archive Network (CRAN) at https://cran.r-project.org/package=fragility. Users can directly install the package by typing `install.packages("fragility")` in R. Once the package is installed, users can begin to use the package by loading it:

```
> library("fragility")
```

The package imports functions from "metafor" [88] for performing pairwise MAs and "netmeta" [89] for performing NMAs. We first introduce example datasets included in "fragility" to demonstrate the data structures, and then provide details about the functions for assessing and visualizing fragility.

**Example datasets.**  The package "fragility" provides four datasets, `dat.ad`, `dat.ns`, `dat.copd`, and `dat.sc`. They all consist of multiple clinical studies, and are used for different illustrative purposes.

The dataset `dat.ad` contains 347 randomized controlled trials of antidepressant drugs with a binary acceptability (dropout due to any cause) outcome; these trials were systematically collected by Cipriani et al. [90]. This dataset is used to illustrate the usage of functions for assessing and visualizing the fragility of individual studies. We display the first six trials as follows:

```
> data("dat.ad")
> head(dat.ad)
  e0  n0 e1   n1
```

```
1  7 107 12 105
2 17 118 18 120
3 30 252 49 263
4 25 109 19 109
5 35 167 35 168
6 17 137 26 140
```

Each row presents the data of a trial. The columns e0, n0, e1, and n1 present event counts and sample sizes in group 0 and those in group 1, respectively. Of note, we use this dataset as an example of (multiple) individual studies, although Cipriani et al. [90] originally performed an NMA based on this dataset. The two treatments (antidepressant drugs or placebo) compared in each study may be different. This dataset does not include multi-arm trials originally collected by Cipriani et al. [90].

The dataset dat.ns contains a collection of 564 pairwise MAs on nutrition support retrieved from Feinberg et al. [91]. Each MA may compare different treatments and have different binary outcomes. This dataset is used to illustrate the usage of functions for assessing and visualizing the fragility of pairwise MAs. Its first six rows are:

```
> data("dat.ns")
> head(dat.ns)
ma.id  e0 n0 e1  n1
1      1  3  24  4 20
2      1  2  10  1 9
3      1  2  28  0 22
4      1 31 265 46 260
5      1  6  32  4 28
6      1  4  35  5 39
```

Each row represents a specific study in a specific MA. The first column ma.id indexes the MAs, ranging from 1 to 564; the output above is from the first six studies in the first MA. The remaining four columns e0, n0, e1, and n1 have the same interpretations as in the dataset dat.ad of individual studies. Some MAs may have overlapping studies, and some may be divided into several subgroups.

Finally, the datasets dat.copd and dat.sc are used to illustrate the usage of functions for assessing and visualizing the fragility of NMAs. The dataset dat.copd is extracted from Woods et al. [92]; it gives a simple NMA with 3 studies comparing 4 treatments for chronic obstructive pulmonary disease. As this dataset is small, the assessment of its fragility does not take much time, and thus it serves as a toy example. The full dataset is:

```
> data("dat.copd")
> dat.copd
  sid tid  e   n
1   1   3 1 229
2   1   1 1 227
3   2   2 4 374
4   2   3 3 372
5   2   4 2 358
6   2   1 7 361
7   3   3 1 554
8   3   1 2 270
```

The data structure of the NMA is different from those of individual studies and pairwise MAs introduced above. Specifically, each row represents the data from a specific treatment group in a specific study. The columns sid and tid give the indexes of studies and

treatments, respectively, and e and n give the corresponding event counts and sample sizes. The four treatments in this dataset are indexed as 1) placebo; 2) fluticasone; 3) salmeterol; and 4) salmeterol fluticasone combination. As shown in the output above, studies 1 and 3 are two-armed, while study 2 is four-armed. In addition to this simple dataset, the package "fragility" also includes a larger NMA dataset of smoking cessation, dat.sc. Its first six rows are displayed as follows:

```
> data("dat.sc")
> head(dat.sc)
  sid tid e    n
1   1   1 9 140
2   1   3 23 140
3   1   4 10 138
4   2   2 11 78
5   2   3 12 85
6   2   4 29 170
```

This dataset is retrieved from Lu and Ades [93] that used formal methods to perform the NMA, while it was originally reported in Hasselblad [94]. It has the same data structure as in dat.copd. The NMA contains 24 studies comparing 4 treatments: 1) no contact; 2) self-help; 3) individual counseling; and 4) group counseling. The binary outcome is successful smoking cessation. The first two studies are three-armed as shown in the output above, and the remaining 22 studies are two-armed.

**Functions for assessing the fragility of individual studies.** Three functions, frag.study(), frag.study.alpha(), and frag.studies(), are available in the package "fragility" to assess the fragility of individual studies. The function frag.study() assesses the fragility of a single study; frag.study.alpha() assesses an individual study's fragility at different significance levels; and frag.studies() assesses the fragility of multiple individual studies.

The arguments of the function frag.study() include:

```
frag.study(e0, n0, e1, n1, data, all = FALSE, methods,
  modify0 = "both", modify1 = "both", alpha = 0.05,
  alternative = "two.sided", OR = 1, RR = 1, RD = 0,
allcase = TRUE)
```

where e0, n0, e1, and n1 specify event counts and sample sizes in groups 0 and 1. The argument data is optional for specifying the dataset; if specified, the previous four arguments should be the corresponding column names in data. The logical argument all indicates whether all possible event status modifications will be considered for assessing the study's fragility. If users only need the numerical value of FI or FQ and the corresponding event status modifications that alter the significance or non-significance, then all = FALSE (the default) is sufficient to produce these results via an iterative algorithm (i.e., starting from modifying one event's status, until the significance or non-significance is altered). The output of this function is of class "frag.study". If all = TRUE, this function generates *p* values corresponding to all possible event status modifications, so that users are able to visualize the extent of significance based on these *p* values. In this case, the output is of both classes "frag.study" and "frag.study.all". The visualization can be easily performed using the function plot() via the S3 method for class "frag.study.all" (detailed later). If the study has large sample sizes (n0 and n1) in both treatment groups and there may be many possible event status modifications, all is recommended to be set to FALSE because R may run out of memory; for example, a study with 1000 samples in each group may lead to up to

one million possible event status modifications. Appendix A in S1 File includes more details of additional arguments.

The function `frag.study.alpha()` efficiently assesses an individual study's fragility at different significance levels, and produces the average FI and FQ across these levels. Its arguments include:

```
frag.study.alpha(e0, n0, e1, n1, data, methods,
  modify0 = "both", modify1 = "both",
  alpha.from = 0.005, alpha.to = 0.05, alpha.breaks = 100,
  alternative = "two.sided", OR = 1, RR = 1, RD = 0)
```

All arguments except the second line are the same as their counterparts in `frag.study()`; the second line specifies the range of possible significance levels, which may be particularly useful if clinicians have different opinions about defining statistical significance [56, 57]. Specifically, `alpha.from`, `alpha.to`, and `alpha.breaks` specify the smallest and largest values of the significance levels to be considered, and the number of levels, respectively. The candidate significance levels are equally spaced within the range. This function produces an object of classes "`frag.alpha`" and "`frag.study.alpha`". The FIs or FQs assessed at different significance levels can be visualized as a step-like function using `plot()` via the S3 method for class "`frag.alpha`" (detailed later).

The function `frag.studies()` permits users to input multiple studies for assessing their fragility. It is particularly useful if users would like to conduct an overall assessment among a collection of studies (e.g., trials belonging to some similar specialties) and investigate the distribution of their fragility measures [31, 32]. Its arguments are similar to those of `frag.study()`; they are displayed as follows:

```
frag.studies(e0, n0, e1, n1, data, methods,
  modify0 = "both", modify1 = "both", alpha = 0.05,
  alternative = "two.sided", OR = 1, RR = 1, RD = 0)
```

All arguments have the same usage as in `frag.study()`, except that `e0`, `n0`, `e1`, and `n1` specify vectors of event counts and sample sizes from the multiple studies, instead of single numerical values. The function output is of classes "`frag.multi`" and "`frag.studies`"; users can apply `plot()` to the output for generating a bar plot or histogram to visualize the overall distribution of the multiple studies' FIs or FQs via the S3 method for class "`frag.multi`".

**Functions for assessing the fragility of pairwise meta-analyses.**   Similar to the three functions above for assessing individual studies' fragility, "fragility" offers `frag.ma()`, `frag.ma.alpha()`, and `frag.mas()` for assessing the fragility of pairwise MAs. The package imports the function `rma.uni()` from "metafor" [88] to perform pairwise MAs and obtain the effect size estimates (including CIs), which further determine the FIs or FQs. Users may refer to [95] for many additional arguments that can be used to customize the MAs.

The major function `frag.ma()` for assessing a pairwise MA's fragility has the following arguments:

```
frag.ma(e0, n0, e1, n1, data, measure = "OR", alpha = 0.05,
  mod.dir = "both", OR = 1, RR = 1, RD = 0, method = "DL", test
  = "z",
  ...)
```

where `e0`, `n0`, `e1`, and `n1` specify the event counts and sample sizes of each study in the MA, and the optional argument `data` can specify the MA dataset. One of the three effect measures, OR, RR, and RD, may be specified for `measure`, and the arguments OR, RR, and RD give the corresponding null values. The argument `alpha` specifies the significance level; it corresponds to the confidence level $(1 - \text{alpha}) \times 100\%$ of CIs. The argument `mod.dir`

indicates the direction of the CI change due to event status modifications when the original MA's CI covers the null value (i.e., the case of non-significance altered to significance). It is not used if the original MA has a significant estimate. Users may specify "left" (moving the CI to the left side of the null value), "right" (moving the CI to the right side), "one" (based on the direction of the original point estimate of the overall effect size), or "both" (both directions) for mod.dir. The default option "both" is expected to find the minimal event status modifications for altering the non-significance, but it may require more computation time than the other three options. Appendix A in S1 File includes more details of additional arguments. The function frag.ma() returns an object of class "frag.ma"; users can apply plot() to the output via the S3 method for this class to visualize the iterative process of event status modifications for deriving the fragility measure of the MA.

The function frag.ma.alpha() assesses the fragility of an MA at multiple significance levels. Its relationship with frag.ma() is similar to that between frag.study() and frag.study.alpha(). Its arguments are the same as frag.ma(), except that users can specify a range of significance levels using the arguments alpha.from, alpha.to, and alpha.breaks. The function returns an object of classes "frag.alpha" and "frag.ma.alpha"; like the output of frag.study.alpha(), it can be visualized using plot() via the S3 method for "frag.alpha".

The function frag.mas() assesses the fragility of multiple MAs; its relationship with frag.ma() is similar to that between frag.study() and frag.studies(). It returns an object of classes "frag.mas" and "frag.multi", and users can visualize the fragility measures among the multiple MAs using plot() via the S3 method for "frag.multi". Its arguments slightly differ from frag.ma():

```
frag.mas(e0, n0, e1, n1, ma.id, data, measure = "OR",
alpha = 0.05,
  mod.dir = "both", OR = 1, RR = 1, RD = 0, method = "DL", test
  = "z",
  ...)
```

The major difference is about the arguments e0, n0, e1, n1, and ma.id for inputting data. Users may refer to the structure of the example dataset dat.ns introduced previously. Specifically, ma.id is a vector for indexing the multiple MAs, and e0, n0, e1, and n1 specify the event counts and sample sizes of each study in each MA. Like frag.ma(), users may specify additional arguments from "metafor" for frag.mas(), as well as frag.ma.alpha(), to customize the implementation of MAs.

**Functions for assessing the fragility of network meta-analyses.**   In addition, "fragility" provides two functions frag.nma() and frag.nma.alpha() for assessing the fragility of NMAs. These are designed for the similar purposes to frag.ma() and frag.ma.alpha(); that is, frag.nma() deals with an NMA at a specific significance level, while frag.nma.alpha() assesses the fragility at multiple significance levels. However, these two functions' arguments may involve more specifications than their counterparts for pairwise MAs, owning to the more complicated structure of NMAs. The functions pairwise() and netmeta() imported from "netmeta" [89] are used to implement NMAs. Of note, "fragility" does not provide a function for simultaneously assessing the fragility of multiple NMAs like frag.studies() and frag.mas(), because a single NMA can be viewed as a comprehensive collection of many pairwise MAs for comparisons of all available treatments. Usually, only a few NMAs are available on certain common topics. In such cases, users may apply frag.nma() to each NMA separately for assessing their overall fragility.

The arguments of frag.nma() are as follows:

```
frag.nma(sid, tid, e, n, data, measure = "OR", random = TRUE,
```

```
alpha = 0.05, mod.dir = "both", tid1.f, tid2.f,
OR = 1, RR = 1, RD = 0,
incr, allincr, addincr, allstudies, ...)
```

where `sid`, `tid`, `e`, and `n` specify study IDs, treatment IDs, their corresponding event counts and sample sizes. One may also specify the dataset for the optional argument `data`. We recommend using the natural numbers (starting from 1) to index the studies and treatments; otherwise, the functions imported from "netmeta" may give warnings that treatments are re-sorted according to a certain order. Moreover, the arguments `measure`, `alpha`, `mod.dir`, `OR`, `RR`, and `RD` have the same usage as in `frag.ma()` for pairwise MAs. The logical argument `random` indicates whether the NMA is performed under the fixed-effects setting (`FALSE`) or random-effects setting (`TRUE`, the default). The two arguments `tid1.f` and `tid2.f` specify the treatment comparison(s) of interest for the assessment of fragility; the default is that the fragility is assessed for all treatment comparisons. For example, if `tid1.f = 1` and `tid2.f = 2`, then the function only assesses the fragility of 1 vs. 2; if `tid1.f = c(2, 3)` and `tid2.f = c(1, 2)`, then it assesses the fragility of 2 vs. 1 and 3 vs. 2. The four arguments `incr`, `allincr`, `addincr`, and `allstudies` are used for handling zero event counts; they are passed to `pairwise()` in "netmeta." Users may additionally specify arguments from `netmeta()` to customize the implementation of the NMAs; see its manual for more details [89]. The output of `frag.nma()` is of class "frag.nma". It can be visualized using `plot()` via the S3 method for class "frag.nma" to show the iterative process of event status modifications for deriving the fragility measure of a specific treatment comparison.

The function `frag.nma.alpha()` assesses the fragility of an NMA at multiple significance levels, similar to `frag.study.alpha()` and `frag.ma.alpha()`. Most arguments are the same as `frag.nma()`, except the arguments `alpha.from`, `alpha.to`, and `alpha.breaks` for specifying the range of candidate significance levels. Because it may be time-consuming to perform many NMAs for deriving the fragility measures, we recommend users to specify a relatively small number of significance levels to `alpha.breaks`, especially for large NMAs. The output of `frag.nma.alpha()` is of classes "frag.alpha" and "frag.nma.alpha"; again, users can use `plot()` via the S3 method for "frag.alpha" to visualize the relationship between fragility measures and significance levels for a specific treatment comparison.

**Summary of data types, functions, and output classes.** Table 2 summarizes the functions and their output classes for each data type. The object produced by each function is a list

**Table 2. Summary of major functions (followed by parentheses) and their output classes (within quotation marks) in the package "fragility" for assessing the fragility of different data types.**

| Data type | Function and output class under each scenario | | |
|---|---|---|---|
| | Single significance level and single dataset | Multiple significance levels and single dataset | Single significance level and multiple datasets |
| Individual study | `frag.study();` | `frag.study.alpha();` | `frag.studies();` |
| | "frag.study" and | "frag.alpha" and | "frag.multi" and |
| | "frag.study.all" (if `all = TRUE`) | "frag.study.alpha" | "frag.studies" |
| Pairwise meta-analysis | `frag.ma()` | `frag.ma.alpha()` | `frag.ma()` |
| | "frag.ma" | "frag.alpha" and | "frag.multi" and |
| | | "frag.ma.alpha" | "frag.mas" |
| Network meta-analysis | `frag.nma()` | `frag.nma.alpha()` | Not applicable |
| | "frag.nma" | "frag.alpha" and | |
| | | "frag.nma.alpha" | |

containing different elements about the input data, relevant estimates, and their fragility measures. It is automatically printed by `print()` via the S3 method for its corresponding class(es). The printed messages are informative summaries of the data, analyses, and assessments of the fragility. If users would like to obtain more comprehensive information, they can extract elements from the output list; the elements' names in the list can be found by applying the function `names()`.

**Functions for visualizing the fragility.** The package "fragility" offers functions for visualizing the fragility of individual studies, pairwise and NMAs; they are called by `plot()` via the S3 method for certain classes.

To visualize the fragility of an individual study, users need to specify `all = TRUE` in `frag.study()` so that all possible event status modifications are considered. The produced object belongs to the class "`frag.study.all`"; for this object, the arguments of the visualization function are as follows:

```
plot(x, method, modify0, modify1, trun, xlab, ylab, xlim,
ylim,
  cex.pts, cex.legend.pval, cex.legend.title,
  col.ori, col.ori.hl, col.f.hl, col.sig, lty.ori, lwd.ori,
  pch, pch.ori, pch.ori.hl, pch.f, pch.f.hl, pch.trun,
  adjust.legend, adjust.seg, legend.pvals, ...)
```

where `x` is the output of `frag.study()` with `all = TRUE`. Users may only specify a single statistical method used to calculate the *p* value for the argument `method` when visualizing the fragility at one time; it must be an element of x$methods, i.e., the argument `methods` specified for `frag.study()`. If `method` is not specified, then the first method in x $methods is used. The arguments `modify0` and `modify1` specify logical values indicating whether event status is modified in groups 0 and 1, respectively, for the visualization. When both `modify0` and `modify1` are `TRUE`, the generated plot presents *p* values (with different colors representing their magnitudes) based on all possible event status modifications; the modifications in groups 0 and 1 are presented on the x and y axes, respectively. A legend is displayed to relate different colors to *p* value magnitudes. When only one of `modify0` and `modify1` is `TRUE`, a scatter plot is generated. It presents *p* values (on a base-10 logarithmic scale) on the y axis against modifications in group 0 (if `modify0 = TRUE`) or group 1 (if `modify1 = TRUE`) on the x axis. The default of `modify0` and `modify1` is `TRUE` if the range of modifications in the corresponding treatment group, which is stored in the object `x` (i.e., x$f0.range or x$f1.range), is not 0; otherwise, the default is `FALSE`. Appendix B in S1 File includes more details of additional arguments.

To visualize the fragility of a pairwise MA, users may apply `plot()` via the S3 method for class "`frag.ma`" to the object x produced by `frag.ma()` as follows:

```
plot(x, xlab, ylab, xlim, ylim, ybreaks = NULL, study.
marker = TRUE,
  cex.marker, offset.marker, col.line, lwd,
  legend, x.legend, y.legend, cex.legend, ...)
```

This generates a plot showing the iterative process of event status modifications, where the x axis presents the iterations and the y axis gives the group-specific total event counts. As the total event counts of the two treatment groups may differ greatly, users may specify a range (a vector of two numerical values) for the argument `ybreaks` to break the y axis for better visualization. The default of this argument is `NULL` (i.e., not breaking the y axis). The specified range should be between the total event counts of the two groups. The axis break is implemented by importing `axis.break()` from "plotrix" [96]. The argument `study.marker` specifies a logical value indicating whether study labels involved in modifications are presented.

When it is `TRUE` (the default), an asterisk represents that the corresponding study with an event status modification remains the same as in the previous iteration. The study labels can be adjusted by the arguments `cex.marker` (text size) and `offset.marker` (distance from lines). The remaining arguments are mainly used to specify certain graphical parameters; again, additional arguments from `plot.default()` can be specified for customizing the plot. A legend is automatically presented to identify the two treatment groups; it can be modified by the last three arguments, which are passed to `legend()` in "graphics." The default is to place the legend on the right side with `x.legend = "right"` and `y.legend = NULL`; in cases that the default legend box overlaps with the lines of the event status modification process, users may specify other coordinates or keywords to change the legend location.

The visualization function for an NMA is similar to the function above for a pairwise MA. Specifically, the arguments of `plot()` via the S3 method for class "`frag.nma`" include:

```
plot(x, tid1, tid2, xlab, ylab, xlim, ylim, ybreaks = NULL,
   study.marker = TRUE, cex.marker, offset.marker, col.line,
   lwd,
   legend, x.legend, y.legend, cex.legend, ...)
```

where `x` is the output from `frag.nma()`. Most arguments are the same with those for class "`frag.ma`" of a pairwise MA. The major difference is regarding the arguments `tid1` and `tid2`, which specify the two treatments of the comparison of interest (i.e., `tid1` vs. `tid2`). Only one comparison can be specified by `tid1` and `tid2` at one time for visualization. If these two arguments are not specified, the first comparison stored in `x$tid.f` is used.

In addition to the three functions above for a single dataset, "fragility" provides two functions for visualizing the relationship between fragility measures and significance levels and for generating overall distributions of fragility measures among multiple datasets. Specifically, for an object x of class "`frag.alpha`" produced by `frag.study.alpha()`, `frag.ma.alpha()`, or `frag.nma.alpha()`, one may visualize it using `plot()` via the S3 method for this class with the following arguments:

```
plot(x, method, fragility = "FI", percentage = TRUE, xlab,
ylab,
   xlim, ylim, cex.pts, col.line, col.pval, col.sig, lty.pval,
   lwd, lwd.pval, pch, pch.inf, tid1, tid2, FQ.nma = FALSE, ...)
```

In the generated plot, the x axis presents the significance levels, and the y axis presents the corresponding fragility measures. Appendix B in S1 File gives details of these arguments.

For an object of class "`frag.multi`", the visualization function is:

```
plot(x, method, dir = "both", fragility = "FI",
percentage = TRUE,
   max.f = NULL, bar, names.arg, space = 0, breaks, freq,
   reverse = FALSE, xlab, ylab, main = NULL, cex.marker, col.
   border,
   col.sig, trun.marker = TRUE, ...)
```

where `x` is the output from `frag.studies()` and `frag.mas()`. This function generates a bar plot or histogram (depending on the specified arguments) to show the overall distribution of fragility measures among the multiple datasets of individual studies or pairwise MAs. Appendix B in S1 File gives details of these arguments and the generated plot.

## Results

This section presents worked examples to illustrate the usage of the various functions in "fragility." These examples are based on the datasets introduced earlier; users may first load them

before implementing the following code. We focus on illustrating the usage of several major arguments for each function with detailed interpretations; users may refer to the manual of "fragility" for more examples that specify many other arguments for various purposes. The results were obtained using R (version 4.0.2) with "fragility" (version 1.1).

## Example of an individual clinical study

Recall that the dataset `dat.ad` consists of 347 trials; each row presents the data of one trial. We first apply the function `frag.study()` to assess the fragility of trial 13; the code and output are in Fig 1. The produced object `out.trial13` is of class "`frag.study`", and the informative output is displayed via the print method for this class. The output consists of three parts. The first part gives the information about the original 2×2 table and the ranges of event status modifications in both groups are presented. The second part displays the information about the significance, including the pre-specified significance level, null value(s) (if using the OR, RR, and/or RD), and the $p$ value(s) with the associated method(s). The third part contains the major information about the fragility, including the FI and FQ based on each method considered, the direction of significance change, and the corresponding minimal event status modification(s) for altering significance or non-significance.

In this example, all arguments besides those receiving data input are set to the default, so all five methods, i.e., Fisher's exact test, the chi-squared test, OR, RR, and RD, are considered. All methods indicate significant results in the original dataset at the level of 0.05, so the directions of their FIs are significance altered to non-significance. All methods except the RD have FIs of 6, while the RD has an FI of 7. The FI may be derived by multiple minimal event status modifications for some methods. As the produced object `out.trial13` is a list, users can apply `names()` to obtain the names of all list elements and thus retrieve the results of interest; they may refer to the manual of "fragility" for details about each element. For example, the FIs of all five methods can be retrieved as follows:

```
> out.trial13$FI
Fisher chisq      OR      RR      RD
    6      6       6       6       7
```

To visualize the fragility of trial 13, users need to specify `all = TRUE` in the `frag.study()` function:

```
> out.trial13.all <- frag.study(e0 = e0, n0 = n0, e1 = e1,
n1 = n1,
      data = dat.ad[13,], all = TRUE)
```

The produced output is also of class "`frag.study.all`"; it can be visualized as follows:

```
> plot(out.trial13.all, method = "Fisher",
    main = "FI = 6, significance altered to non-significance",
    cex.pts = 0.5, font.main = 1, cex.main = 0.9)
> plot(out.trial13.all, method = "Fisher", modify1 = FALSE,
    main = "FI = 6, significance altered to non-significance",
    cex.pts = 0.8, font.main = 1, cex.main = 0.9)
> plot(out.trial13.all, method = "Fisher", modify0 = FALSE,
    main = "FI = 7, significance altered to non-significance",
    cex.pts = 0.8, font.main = 1, cex.main = 0.9)
```

Here, Fisher's exact test is used to calculate $p$ values. Fig 2a–2c present the generated plots.

The first function implementation in the code above visualizes the fragility of trial 13 by modifying event status in both treatment groups; the argument `cex.pts` specifies the size of points in Fig 2a. Each point represents a $p$ value based on certain event status modifications

```
> out.trial13 <- frag.study(e0 = e0, n0 = n0, e1 = e1, n1 = n1, data = dat.ad[13,])
> out.trial13
_____________________________________
Original data:
        event no event
group 0    16      149
group 1    36      131
Range of event modification in group 0:
  up to 16 events modified to be non-events;
  up to 149 non-events modified to be events
Range of event modification in group 1:
  up to 36 events modified to be non-events;
  up to 131 non-events modified to be events
_____________________________________
Significance level = 0.05
Null hypothesis: OR = 1, RR = 1, RD = 0
p-value (two-sided):
  0.004 based on Fisher's exact test
  0.005 based on chi-squared test
  0.004 based on odds ratio
  0.004 based on relative risk
  0.003 based on risk difference
_____________________________________
Fragility index (FI) and fragility quotient (FQ):
Based on Fisher's exact test, FI = 6 (FQ = 1.8%)
  for significance altered to non-significance,
  achieved by inversing status of
  6 non-events in group 0; or
  4 non-events in group 0 and 2 events in group 1; or
  5 non-events in group 0 and 1 event in group 1
Based on chi-squared test, FI = 6 (FQ = 1.8%)
  for significance altered to non-significance,
  achieved by inversing status of
  6 non-events in group 0; or
  2 non-events in group 0 and 4 events in group 1; or
  3 non-events in group 0 and 3 events in group 1; or
  4 non-events in group 0 and 2 events in group 1; or
  5 non-events in group 0 and 1 event in group 1
Based on odds ratio, FI = 6 (FQ = 1.8%)
  for significance altered to non-significance,
  achieved by inversing status of
  6 non-events in group 0
Based on relative risk, FI = 6 (FQ = 1.8%)
  for significance altered to non-significance,
  achieved by inversing status of
  6 non-events in group 0
Based on risk difference, FI = 7 (FQ = 2.1%)
  for significance altered to non-significance,
  achieved by inversing status of
  7 non-events in group 0; or
  3 non-events in group 0 and 4 events in group 1; or
  4 non-events in group 0 and 3 events in group 1; or
  5 non-events in group 0 and 2 events in group 1; or
  6 non-events in group 0 and 1 event in group 1
```

**Fig 1. Output from the function `frag.study()`.**

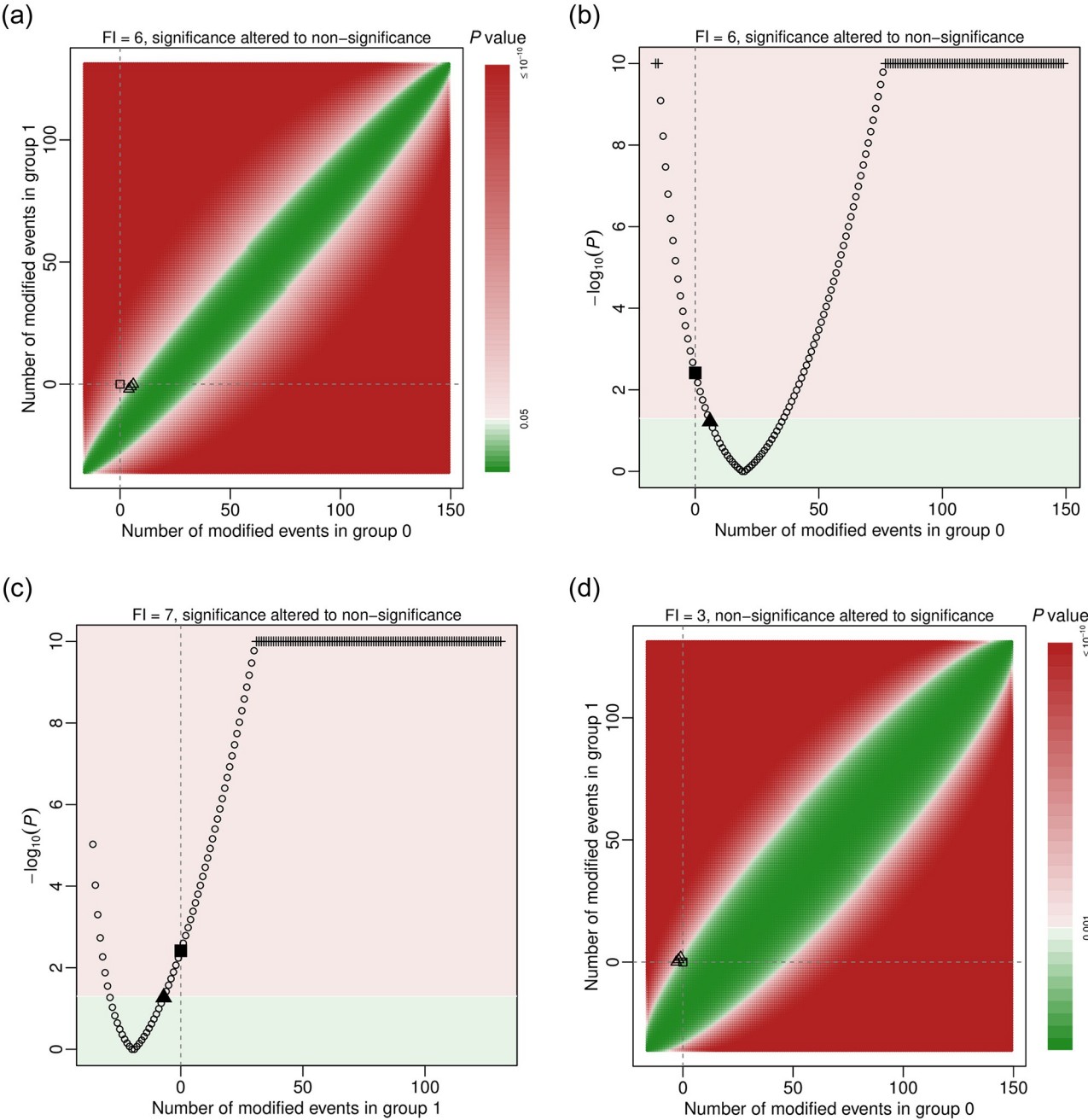

**Fig 2. Visualizations of trial 13 in the dataset `dat.ad`.** Points or areas in green indicate non-significant results, and those in red indicate significant ones. Dashed lines represent no modifications in the corresponding groups. Square points represent the original *p* value, and triangle points indicate minimal modifications that alter the significance. In panels (a) and (d), event statuses are modified in both groups. Each point represents the extent of the *p* value corresponding to a specific combination of event status modifications in groups 0 and 1. In panels (c) and (d), event statuses are modified in only one group. Plus points represent truncated *p* values at $10^{-10}$.

given by the x axis (group 0) and the y axis (group 1). By default, the significance level is 0.05; the *p* values lower than this level (significant results) are presented in red, and those above this level (non-significant results) are in green. The legend on the right side indicates the magnitudes of *p* values; the color opacity of a *p* value changes linearly according to the negative base-

10 logarithm of the *p* value. Because `trun` is set to 10 by default, *p* values lower than $10^{-10}$ are truncated. The non-significant results are generally around a diagonal line, where the event status is modified so that the even counts in the two groups are close, leading to large *p* values. On the other hand, for points away from the diagonal line, the difference between the modified event counts in the two groups becomes larger, so the corresponding *p* values are smaller. In addition, the vertical and horizontal dashed lines indicate no modifications in groups 0 and 1, respectively; they cross at a square point, corresponding to the *p* value of the original data. This *p* value is located in the red area, implying a significant result; therefore, to assess the fragility of this trial, we aim at modifying event statuses so that the original *p* value is moved to the green area of non-significant results. The three triangle points in the green area indicate three cases of minimal event status modifications that can alter the significance to non-significance. They represent 1) changing 6 non-events in group 0 to events; 2) changing 5 non-events in group 0 to events and 1 event to a non-event in group 1; and 3) changing 4 non-events in group 0 to events and 2 events to non-events in group 1. These match the output of `out.trial13` displayed previously. All three cases indicate FI = 6.

The second function implementation sets `modify1 = FALSE` to visualize the fragility by restricting the modifications to group 0 (Fig 2b). As event status is only modified in group 0, this plot presents the negative base-10 logarithm of *p* values against the corresponding modifications. The *p* values in this plot correspond to those on the horizontal dashed line at 0 in Fig 2a. The red area at the top indicates significant results, and the green area at the bottom indicates non-significant results. Again, the *p* values lower than $10^{-10}$ are truncated; the truncated *p* values are presented as plus signs. The vertical dashed line at 0 implies the original *p* value (presented as a square point), which is within the red area of significant results. The triangle point represents the minimal event status modification in group 0 for altering the significance to non-significance; it also implies that the FI is 6 when restricting the modifications to group 0 (by changing 6 non-events to events). The numerical value of this FI can be also obtained from the output of `frag.study()` with its argument `all = TRUE`, i.e., `out.trial13.all$FI0`.

Similarly, the third function implementation sets `modify0 = FALSE`; it visualizes the fragility by restricting the modifications to group 1 (Fig 2c). The FI is 7 with this restriction (by changing 7 events to non-events). The numerical value of this FI can be obtained from `out.trial13.all$FI1`.

If users would like to reduce the type I error rate by lowering the significance level $\alpha$ to 0.001, they may simply specify this level for the argument `alpha`:

```
> out.trial13.all.2 <- frag.study(e0 = e0, n0 = n0, e1 = e1,
n1 = n1,
    data = dat.ad[13,], all = TRUE, alpha = 0.001)
> plot(out.trial13.all.2, method = "Fisher",
    main = "FI = 3, non-significance altered to significance",
    cex.pts = 0.5, font.main = 1, cex.main = 0.9)
```

Fig 2d shows the generated plot. Compared with Fig 2a at $\alpha = 0.05$, the original result is no longer significant, and the original *p* value is now within the green area. As the significance level decreases, the green area of non-significant results becomes wider. The original *p* value is close to the border of the green area, implying that this result might be fragile; indeed, the FI becomes 3, and its direction is the non-significance altered to significance. This can be achieved by 1) changing 3 events to non-events in group 0; 2) 2 events to non-events in group 0 and 1 non-event to an event in group 1; or 3) 1 event to a non-event in group 0 and 2 non-events to events in group 1.

```
> out.ma1 <- frag.ma(e0, n0, e1, n1, data = dat.ns[dat.ns$ma.id == 1,])
> out.ma1
Original meta-analysis contains
  99 studies;
  885 total events and 10,153 total sample sizes in group 0;
  831 total events and 10,407 total sample sizes in group 1
Significance level = 0.05
The effect size is OR (on a logarithmic scale)
The null value of is 0
The estimated overall effect size is
  -0.074 with CI (-0.178, 0.030) and p-value 0.165
Fragility index (FI) = 14 and fragility quotient (FQ) = 0.1%
  for non-significance altered to significance
```

**Fig 3. Output from the function `frag.ma()`.**

## Example of a pairwise meta-analysis

We use the dataset dat.ns to illustrate the assessment of the fragility of pairwise MAs. Recall that this dataset contains 564 pairwise MAs on nutrition support. We apply the function frag.ma() to the first MA that investigates the overall all-cause mortality; see Fig 3. All arguments besides those receiving data input are set to the default; that is, the effect measure is the OR with the null value at 1, the significance level is 0.05, the meta-analysis is performed via the DL method, and the CI of the overall effect size is derived based on the normal distribution. The OR is analyzed on a logarithmic scale; the null value of the log OR is 0. The informative output gives a summary of the original data, the evaluation of significance, and the assessment of the fragility. In this example, the CI of the overall log OR of the original data covers 0, indicating a non-significant effect of nutrition support on all-cause mortality. The FI is 14 for altering the non-significance to significance, and the FQ is 0.1%. Due to the space limit, the output does not provide complete results. The produced object out.ma1 is a list that contains many results produced during the iterative process of deriving the FI, including the study and treatment group that are involved in each event status modification, the estimated overall effect size with its CI in each iteration, as well as the data with modified event status in the final iteration where the non-significance is just altered. Users may apply names() to obtain the names of all elements of the produced object.

The package "fragility" does not provide functions to produce classic plots for the pairwise MA, such as the forest plot and funnel plot, because many existing popular packages such as "metafor" [88] and "meta" [97] have included these features. Nevertheless, the process of deriving the FI can be visualized as follows:

```
> plot(out.ma1, ybreaks = c(840, 880), font.main = 1, cex.
main = 0.9,
    main = "FI = 14, non-significance altered to
    significance")
```

Fig 4 presents the produced plot, which shows the total event counts in the two treatment groups during the iterations. It contains two lines that depict the process, where the blue and red lines represent groups 0 and 1, respectively. As the argument ybreaks is specified as c (840, 880), the plot omits this range on the y axis for better visualization. The numbers around the blue line indicate the studies that are involved in the event status modifications during the iterations. Each asterisk indicates that a study remains unchanged as in the previous iteration; that is, the first asterisk represents study 43, and the second represents study 45. No event status is modified in group 1 for deriving the FI in this example.

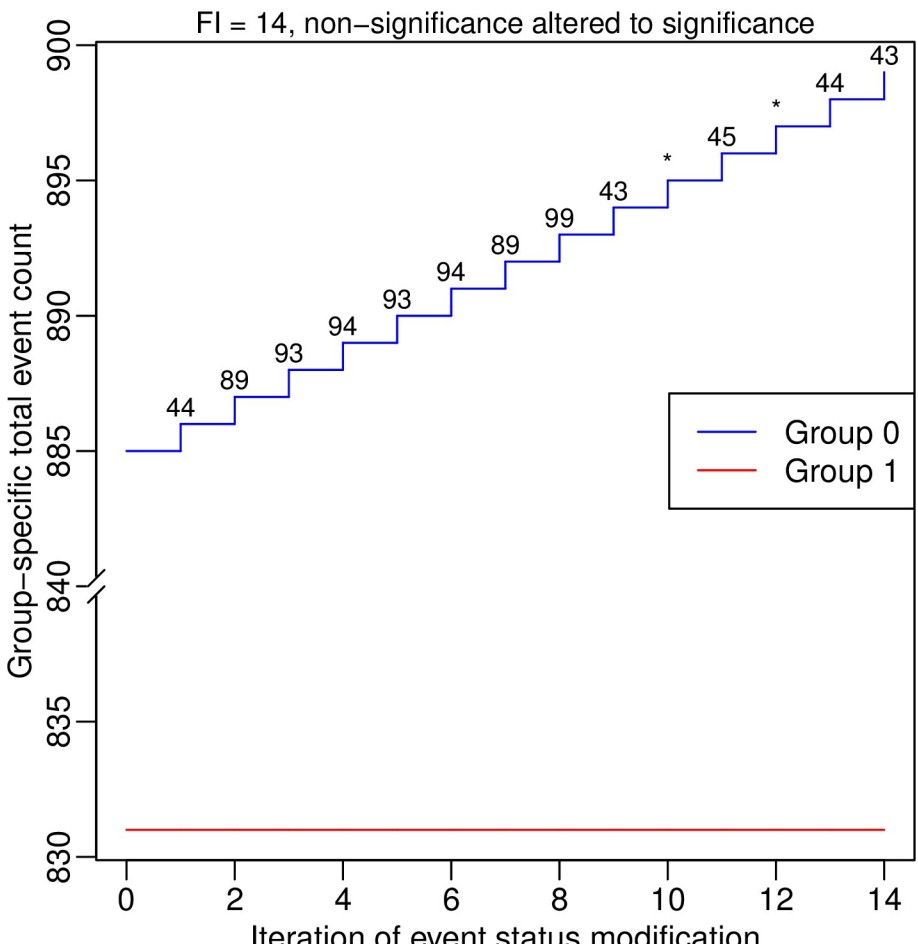

**Fig 4. Event status modifications in the first pairwise meta-analysis in the dataset `dat.ns`.** Each line represents changes in the total event count in a certain group. The numbers around each line indicate the studies involved in the event status modifications during the iterations; each asterisk indicates that a study remains unchanged as in the previous iteration.

### Example of a network meta-analysis

The function `frag.nma()` assesses the fragility of an NMA. We apply it to the dataset `dat.sc` of the NMA on smoking cessation, which contains 24 studies comparing a total of 4 treatments; see Fig 5. We do not specify the arguments `tid1.f` and `tid2.f`, so the fragility of each treatment comparison is assessed. Because many NMAs need to be performed during the iterative algorithm for each comparison, the computation time is around 1 hour; the actual time depends on users' processors.

This NMA contains 4 treatments, so the results of FIs and FQs are presented in 4×4 matrices. The informative output only displays some important characteristics of the NMA and results about fragility. As in the previous examples, more detailed results (e.g., directions of the altered significance or non-significance, studies and treatment groups involved in event status modifications) can be retrieved from the elements of the produced object `out.nma`, whose names can be obtained via applying `names()`. In this example, the FI is as small as 3 for the comparison 4 vs. 1 (a relatively fragile comparison), and it is as large as 32 for 3 vs. 1 (a less fragile comparison).

```
> out.nma <- frag.nma(sid, tid, e, n, data = dat.sc)
> out.nma
Original network meta-analysis (NMA) contains
  24 studies and 4 treatments
Significance level = 0.05
The effect size is OR (on a logarithmic scale)
The null value of is 0
Fragility index (FI):
    1  2  3  4
1 NA 18 32  3
2 18 NA 19 12
3 32 19 NA 23
4  3 12 23 NA
Fragility quotient (FQ), based on the associated comparison:
              1          2          3          4
1          NA 0.002045687 0.002189681 0.000385307
2 0.002045687          NA 0.002122668 0.005652379
3 0.002189681 0.002122668          NA 0.002897455
4 0.000385307 0.005652379 0.002897455          NA
Fragility quotient (FQ), based on the total sample size in the NMA:
              1          2          3          4
1          NA 0.0010754616 0.001911932 0.0001792436
2 0.0010754616          NA 0.001135209 0.0007169744
3 0.0019119316 0.0011352094          NA 0.0013742009
4 0.0001792436 0.0007169744 0.001374201          NA
See the manual for details to retrieve more information.
```

**Fig 5. Output from the function `frag.nma()`.**

The visualization of the process of deriving the FI in an NMA is similar to that in a pairwise MA. The major difference is that the visualization in the NMA needs to be implemented for each treatment comparison separately. We apply `plot()` to the produced object `out.nma` that is of class "frag.nma":

```
> plot(out.nma, tid1 = 2, tid2 = 1, ybreaks = c(170, 595),
    x.legend = "topright", font.main = 1, cex.main = 0.9,
    main = "FI = 18, non-significance altered to
    significance")
> plot(out.nma, tid1 = 3, tid2 = 1, ybreaks = c(635, 1200),
    x.legend = "bottomright", font.main = 1, cex.main = 0.9,
    main = "FI = 32, significance altered to non-
    significance")
> plot(out.nma, tid1 = 4, tid2 = 1, ybreaks = c(105, 600),
    font.main = 1, cex.main = 0.9,
    main = "FI = 3, significance altered to non-significance")
> plot(out.nma, tid1 = 3, tid2 = 2, ybreaks = c(160, 1205),
    font.main = 1, cex.main = 0.9,
    main = "FI = 19, non-significance altered to
    significance")
> plot(out.nma, tid1 = 4, tid2 = 2, ybreaks = c(110, 140),
    x.legend = "topright", font.main = 1, cex.main = 0.9,
    main = "FI = 12, non-significance altered to
    significance")
> plot(out.nma, tid1 = 4, tid2 = 3, ybreaks = c(130, 1205),
```

```
x.legend = "bottomright", font.main = 1, cex.main = 0.9,
main = "FI = 23, non-significance altered to
significance")
```

Fig 6 presents the produced plots. The argument `ybreaks` is specified differently for each comparison because the ranges of the involved total event counts differ. Again, an asterisk represents that a study with modified event status remains unchanged as in the previous iteration. These plots indicate event status is generally modified in a few studies to alter the significance or non-significance. For example, to derive the FI of the comparison 2 vs. 1 in Fig 6a, only studies 2, 16, and 22 among the 24 studies are involved in event status modifications.

The function `frag.nma()` can be similarly applied to the dataset `dat.copd`. This dataset serves as a toy example; its fragility can be assessed much faster due to its small size. Its results are not presented in this article. If an NMA contains many treatments, the assessment of its fragility may take a long time. In this case, users are recommended to only assess the fragility of certain treatment comparisons of primary interest by specifying `tid1.f` and `tid2.f`. Moreover, "fragility" does not provide functions to visualize the NMA, such as the treatment network plot and treatment rank plot, because many existing packages such as "gemtc" [98], "netmeta" [89], and "pcnetmeta" [99] have included these features.

## Example of assessing the fragility at multiple significance levels

The previous examples present the assessment and visualization of the fragility of individual studies, pairwise MAs, and NMAs at a specific significance level. As there are ongoing debates on the choice of statistical significance level [56, 57], users might want to assess the fragility at multiple significance levels. They may apply the functions `frag.study.alpha()`, `frag.ma.alpha()`, and `frag.nma.alpha()` to individual studies, pairwise MAs, and NMAs, respectively, for such purposes. Their usage is similar to their counterparts `frag.study()`, `frag.ma()`, and `frag.nma()`. The produced objects are all of class "frag.alpha", which can be visualized using `plot()` via the S3 method for this class. We focus on an example of an individual study; the code can be similarly applied to pairwise MAs and NMAs.

We continue to use trial 13 in the dataset `dat.ad` for illustrating `frag.study.alpha()`; see Fig 7. The default options are used to specify the range of significance levels, i.e., 100 equally-spaced values between 0.005 and 0.05. The *p* values are derived based on all five methods, i.e., Fisher's exact test, the chi-squared test, OR, RR, and RD. Like the output produced by `frag.study()`, the informative output displays summaries in three parts, which are about original data, significance tests, and fragility. Compared with the output produced by `frag.study()`, the major difference is in the last part about fragility; the output of `frag.study.alpha()` gives the average fragility measures in the range of specified significance levels. The produced object `out.trial13.alpha` is a list, and users can retrieve more detailed information, such as the FI and FQ at each significance level, from this list.

The results can be visualized via `plot()` as follows:

```
> plot(out.trial13.alpha)
> plot(out.trial13.alpha, fragility = "FQ")
```

In the first line, the fragility measure is the FI by default; in the second line, the fragility measure is the FQ. Fig 8a and 8b present the generated plots. As the argument `method` is not specified, the plots are based on the default option, i.e., Fisher's exact test. Because the FQ is the FI divided by the total sample size in the study, which is a constant, the two plots have the same shape; they only differ with respect to the scale on the y axis. Because the FIs must be integers, the plots appear to be step functions. All points in the plots are in red, indicating that the original results are significant at all levels, and the FIs and FQs represent that the

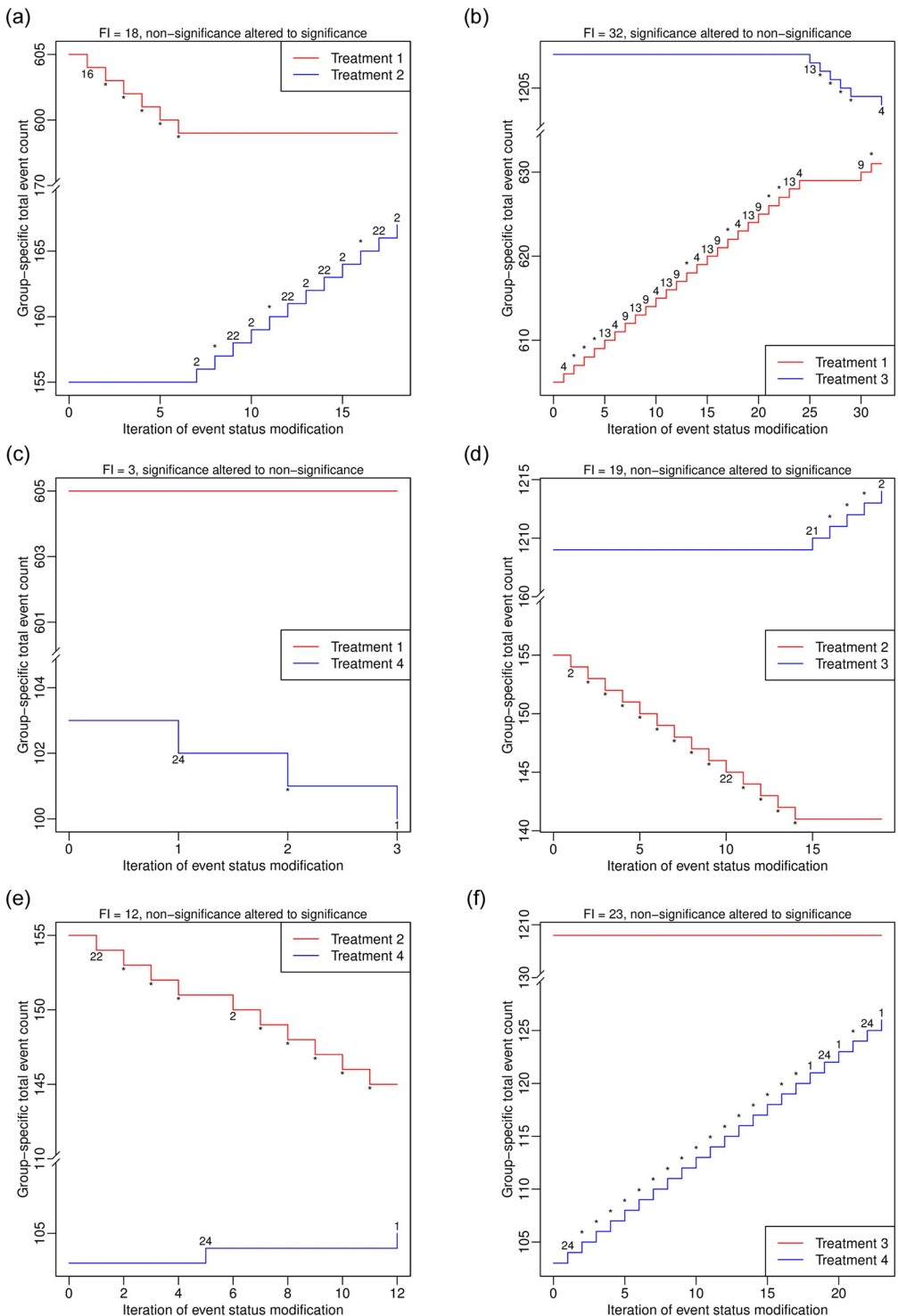

**Fig 6. Event status modifications in the network meta-analysis dataset `dat.sc`.** Each line represents changes in the total event count in a certain group. The numbers around each line indicate the studies involved in the event status modifications during the iterations; each asterisk indicates that a study remains unchanged as in the previous iteration. (a) Comparison 2 vs. 1. (b) Comparison 3 vs. 1. (c) Comparison 4 vs. 1. (d) Comparison 3 vs. 2. (e) Comparison 4 vs. 2. (f) Comparison 4 vs. 3.

```
> out.trial13.alpha <- frag.study.alpha(e0, n0, e1, n1, data = dat.ad[13,])
> out.trial13.alpha
___________________________________________
Original data:
        event no event
group 0    16      149
group 1    36      131
Range of event modification in group 0:
  up to 16 events modified to be non-events;
  up to 149 non-events modified to be events
Range of event modification in group 1:
  up to 36 events modified to be non-events;
  up to 131 non-events modified to be events
___________________________________________
Significance level varies from 0.005 to 0.05
Null hypothesis: OR = 1, RR = 1, RD = 0
p-value (two-sided):
  0.004 based on Fisher's exact test
  0.005 based on chi-squared test
  0.004 based on odds ratio
  0.004 based on relative risk
  0.003 based on risk difference
___________________________________________
Fragility index (FI) and fragility quotient (FQ):
Based on Fisher's exact test,
  Average FI = 4.23 (min = 1, max = 6);
  Average FQ = 1.3% (min = 0.3%, max = 1.8%)
Based on chi-squared test,
  Average FI = 3.88 (min = 1, max = 6);
  Average FQ = 1.2% (min = 0.3%, max = 1.8%)
Based on odds ratio,
  Average FI = 4.56 (min = 1, max = 6);
  Average FQ = 1.4% (min = 0.3%, max = 1.8%)
Based on relative risk,
  Average FI = 4.43 (min = 1, max = 6);
  Average FQ = 1.3% (min = 0.3%, max = 1.8%)
Based on risk difference,
  Average FI = 4.88 (min = 2, max = 7);
  Average FQ = 1.5% (min = 0.6%, max = 2.1%)
```

**Fig 7. Output from the function `frag.study.alpha()`.**

significance is altered to non-significance. As the significance level increases from 0.005 to 0.05, the FI increases from 1 to 6. These correspond to the previous output of `out.trial13.alpha`, and the average FI is 4.23.

Users may specify additional arguments; for example, we change the code to:

```
> out.trial13.alpha.2 <- frag.study.alpha(e0, n0, e1, n1,
    data = dat.ad[13,], alpha.from = 0.001, alpha.to = 0.1,
    alpha.breaks = 500)
> plot(out.trial13.alpha.2)
> plot(out.trial13.alpha.2, log = "x")
```

The significance levels range from 0.001 to 0.1; 500 equally-spaced values are chosen within this range; the results are visualized in Fig 8c. Other arguments from `plot.default()` can be imported; here, we specify `log = "x"` to present the significance levels on a logarithmic scale as in Fig 8d. From the previous output of `out.trial13.alpha`, the *p* value of the

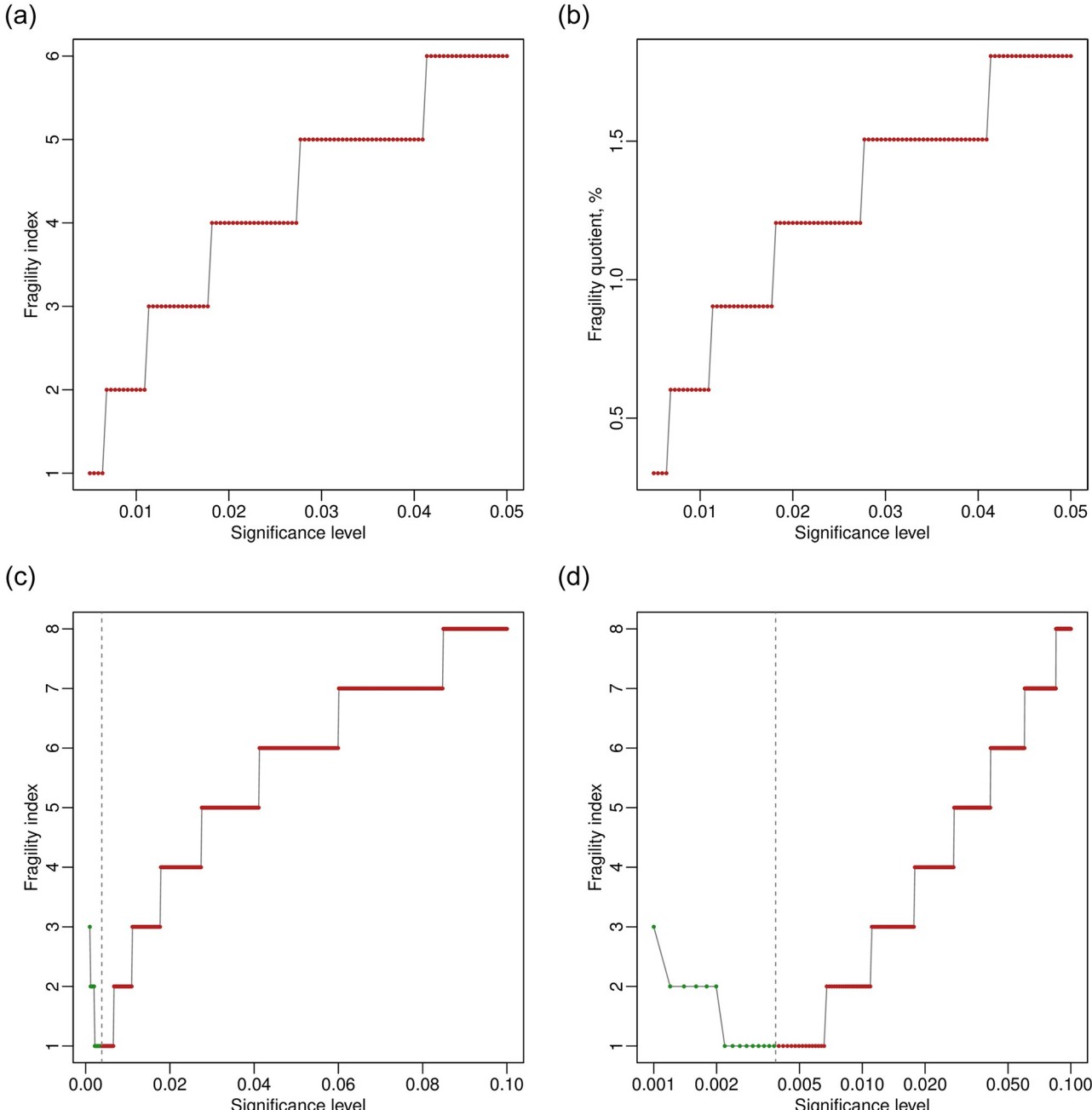

**Fig 8. Visualizations of trial 13 in the `dat.ad` dataset at multiple significance levels.** (a) Fragility index at α from 0.005 to 0.05. (b) Fragility quotient at α from 0.005 to 0.05. (c) Fragility index at α from 0.001 to 0.1. (d) Fragility index at α from 0.001 to 0.1 with the x axis on a logarithmic scale.

original data based on Fisher's exact test is 0.004, so the result is significant if the significance level is above 0.004 but is non-significant if the level is below 0.004. The vertical dashed line in Fig 8c indicates the original *p* value; the FIs on its left side (points in green) represent the non-significance altered to significance, and those on its right side (points in red) represent the significance altered to non-significance. As the significance level increases from 0.001 to 0.1, the FI first decreases from 3 to 1 and then increases from 1 to 8.

## Example of assessing the fragility of multiple datasets

As multiple clinical studies or pairwise MAs (e.g., with different disease outcomes) may be available on certain common topics, clinicians may be interested in the overall distributions of the fragility measures of these studies or pairwise MAs. The functions `frag.studies()` and `frag.mas()` can be used for such purposes. Such a function is not provided for NMAs in "fragility," because usually only a few NMAs are available on common topics. The usage of `frag.studies()` and `frag.mas()` is similar to that of `frag.study()` and `frag.ma()`, respectively. The produced objects of both functions are of class "`frag.multi`"; they can be visualized using `plot()` via the S3 method for this class.

Specifically, we can assess the fragility of all trials contained in the dataset `dat.ad` as in Fig 9. By default, all five methods (Fisher's exact test, the chi-squared test, OR, RR, and RD) are used to derive the fragility measures. The informative output displays a summary of the original data, significance tests, and fragility measures (e.g., medians, ranges, and interquartile ranges [IQRs]). When presenting the fragility measures, the 347 trials are distinguished into two groups, i.e., those with originally significant results and non-significant ones. Users can retrieve complete results from the elements of the output list `out.trials`; for example, the FIs of all trials are stored in `out.trials$FI`.

The fragility measures of all trials can be visualized as follows:

```
> plot(out.trials, method = "Fisher", cex.name = 0.6)
> plot(out.trials, method = "Fisher", max.f = 16, cex.
name = 0.6)
> plot(out.trials, dir = "sig2nonsig", method = "Fisher",
    cex.name = 0.6)
> plot(out.trials, dir = "nonsig2sig", method = "Fisher",
    cex.name = 0.6)
> plot(out.trials, method = "Fisher", fragility = "FQ", max.
f = 20)
> plot(out.trials, method = "Fisher", fragility = "FQ", max.
f = 20,
    breaks = 20)
```

Six plots are produced for different illustrative purposes, as shown in Fig 10. They are based on Fisher's exact test. In the first four function implementations, the argument `fragility` uses the default option, i.e., "`FI`", and `bar` is `TRUE` by default, so Fig 10a–10d present bar plots of FIs. The argument `cex.name` is passed to `barplot()` for adjusting the text size on the x axis; if the size is too large, many values may disappear due to space limit. Fig 10a presents the overall distribution of FIs of all 347 trials. The FIs range from 1 to 19; many trials have FIs between 1 and 10, implying that their significance could be altered by changing the event statuses of at most 10 patients. The bars in red represent trials with originally significant results, so their FIs indicate the significance altered to non-significance; the bars in green represent trials with originally non-significant results, which are altered to be significant. Most trials originally have non-significant results. The FIs of some trials have extreme values, which may affect the visualization effect of the overall distribution. As in the second function implementation of the code above, users can specify `max.f` to truncate FIs above the specified value. Fig 10b presents the overall distribution with FIs truncated at 16; all trials with FIs above 16 are stacked at the rightmost bar. If users want to focus on the direction of FIs that alter the significance to non-significance or its inverse, `dir` can be specified as "`sig2nonsig`" or "`nonsig2sig`", leading to the bar plots in Fig 10c and 10d, respectively. By default, this argument is "`both`", i.e., both directions are presented as in Fig 10a and 10b.

```
> out.trials <- frag.studies(e0, n0, e1, n1, data = dat.ad)
> out.trials
The input dataset contains 347 studies
Significance level = 0.05
Null hypothesis: OR = 1, RR = 1, RD = 0
p-value (two-sided) is based on:
  Fisher's exact test
  chi-squared test
  odds ratio
  relative risk
  risk difference

Fragility index (FI) and fragility quotient (FQ):
Based on Fisher's exact test,
  32 studies yield significance with
    median FI = 3, range 1-13, IQR 1-6 and
    median FQ = 2.1%, range 0.2%-11.1%, IQR 0.7%-4.3%;
  315 studies yield non-significance with
    median FI = 6, range 1-19, IQR 4-8 and
    median FQ = 4.4%, range 0.3%-50.0%, IQR 2.8%-7.2%;
  overall, among all studies,
    median FI = 6, range 1-19, IQR 3-8 and
    median FQ = 4.2%, range 0.2%-50.0%, IQR 2.5%-6.8%
Based on chi-squared test,
  29 studies yield significance with
    median FI = 2, range 1-13, IQR 1-6 and
    median FQ = 1.9%, range 0.1%-8.9%, IQR 1.2%-3.8%;
  318 studies yield non-significance with
    median FI = 6, range 1-19, IQR 4-9 and
    median FQ = 4.6%, range 0.3%-50.0%, IQR 2.9%-7.5%
    while 1 study has FI = FQ = NA;
  overall, among all studies,
    median FI = 6, range 1-19, IQR 4-9 and
    median FQ = 4.4%, range 0.1%-50.0%, IQR 2.8%-7.1%
    while 1 study has FI = FQ = NA
Based on odds ratio,
  38 studies yield significance with
    median FI = 2, range 1-14, IQR 1-5 and
    median FQ = 2.0%, range 0.3%-11.1%, IQR 0.6%-4.3%;
  309 studies yield non-significance with
    median FI = 6, range 1-18, IQR 4-8 and
    median FQ = 4.2%, range 0.3%-50.0%, IQR 2.7%-7.2%
    while 1 study has FI = FQ = NA;
  overall, among all studies,
    median FI = 6, range 1-18, IQR 3-8 and
    median FQ = 4.0%, range 0.3%-50.0%, IQR 2.4%-7.0%
    while 1 study has FI = FQ = NA
Based on relative risk,
  36 studies yield significance with
    median FI = 2, range 1-14, IQR 1-5 and
    median FQ = 2.0%, range 0.2%-11.1%, IQR 0.5%-3.8%;
  311 studies yield non-significance with
    median FI = 6, range 1-19, IQR 4-8 and
    median FQ = 4.3%, range 0.3%-36.8%, IQR 2.8%-7.4%
    while 2 studies have FI = FQ = NA;
  overall, among all studies,
    median FI = 6, range 1-19, IQR 4-8 and
    median FQ = 4.1%, range 0.2%-36.8%, IQR 2.5%-7.2%
    while 2 studies have FI = FQ = NA
Based on risk difference,
  42 studies yield significance with
    median FI = 3, range 1-14, IQR 1-5 and
    median FQ = 2.2%, range 0.3%-16.7%, IQR 0.5%-4.0%;
  305 studies yield non-significance with
    median FI = 5, range 1-18, IQR 3-8 and
    median FQ = 4.1%, range 0.3%-28.6%, IQR 2.5%-6.2%;
  overall, among all studies,
    median FI = 5, range 1-18, IQR 3-8 and
    median FQ = 3.8%, range 0.3%-28.6%, IQR 2.3%-6.1%
```

**Fig 9. Output from the function `frag.studies()`.**

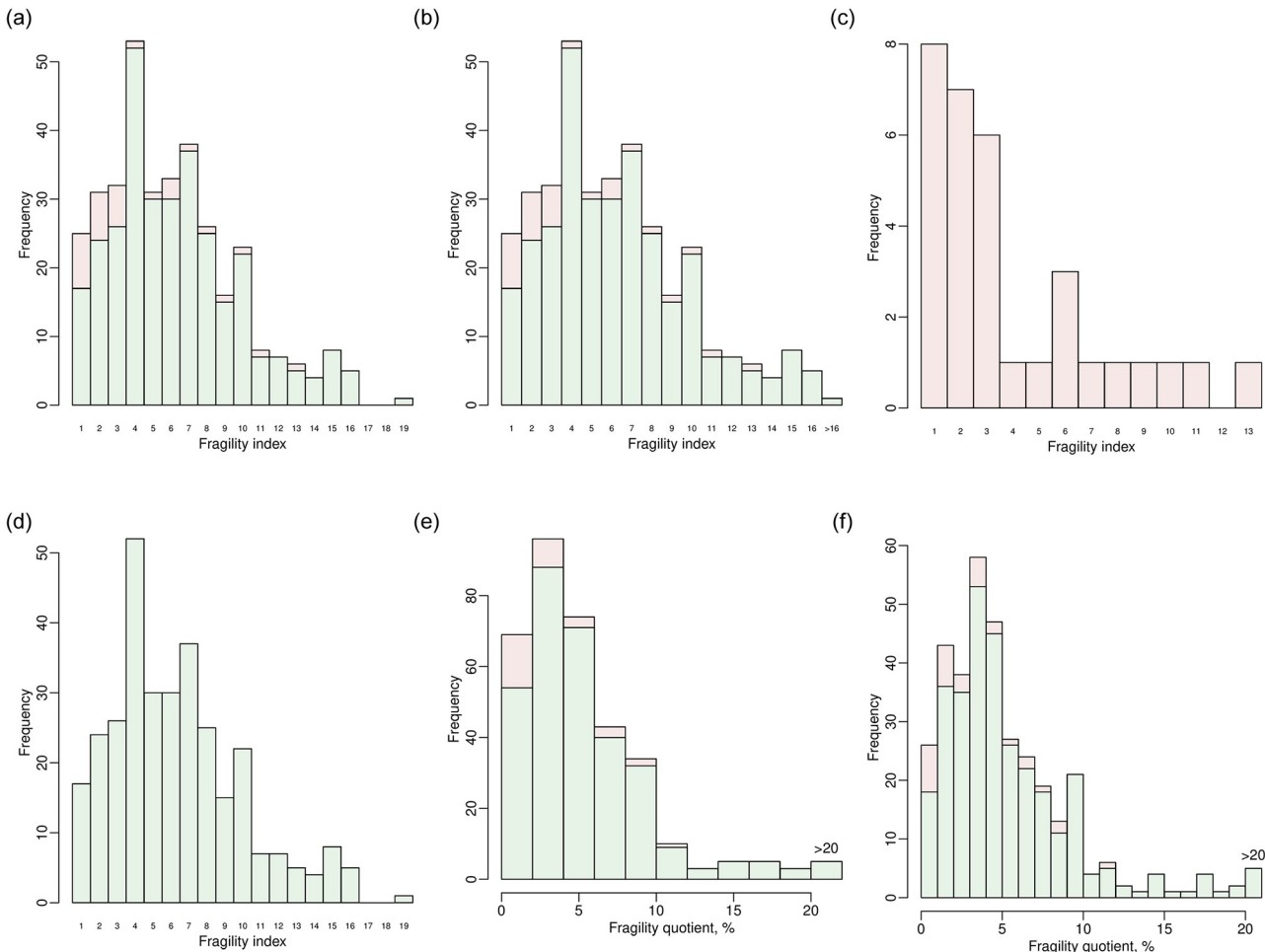

**Fig 10. Distributions of fragility measures of the clinical trials in the dataset `dat.ad`.** (a) Bar plot of fragility indexes of all trials. (b) Bar plot fragility indexes of all trials truncated at 16. (c) Bar plot of fragility indexes of trials whose significance altered to non-significance. (d) Bar plot of fragility indexes of trials whose non-significance altered to significance. (e) Histogram of fragility quotients of all trials. (f) Histogram of fragility quotients of all trials with more breaks.

Alternatively, users can specify `fragility = "FQ"` to produce plots for FQs as in the last two function implementations in the code above. As FQs can take any values within 0%–100%, instead of only integers like FIs, the histogram rather than the bar plot is produced for FQs. Fig 10e presents the overall distribution of FQs, truncated at 20%. If `breaks` is not specified, the number of breaks in the histogram is automatically determined by `hist()`. Users may adjust this argument to change the number of breaks as in Fig 10f.

We also apply `frag.mas()` to the dataset `dat.ns` to assess the fragility of multiple pairwise MAs, as in Fig 11. The effect measure of these MAs is the OR (`measure = "OR"`) by default. The output is similar to that of `out.trials`. It displays a summary of the input MAs, information about significance, and fragility measures. Among the total of 564 pairwise MAs, 97 have significant overall ORs, and their FIs range from 1 to 167; 467 have non-significant overall ORs with FIs ranging from 1 to 61.

The produced object `out.mas` is of class "`frag.multi`", and can be visualized as follows:

```
> out.mas <- frag.mas(e0, n0, e1, n1, ma.id, data = dat.ns)
> out.mas
The input dataset contains 564 meta-analyses
Significance level = 0.05
The effect size is OR (on a logarithmic scale)
The null value of is 0

Fragility index (FI) and fragility quotient (FQ):
  97 meta-analyses yield significance with
    median FI = 11, range 1-167, IQR 5-26 and
    median FQ = 0.2%, range 0.0%-1.4%, IQR 0.1%-0.3%;
  467 meta-analyses yield non-significance with
    median FI = 8, range 1-61, IQR 5-16 and
    median FQ = 0.6%, range 0.0%-8.9%, IQR 0.2%-2.4%;
  overall, among all meta-analyses,
    median FI = 9, range 1-167, IQR 5-17 and
    median FQ = 0.5%, range 0.0%-8.9%, IQR 0.2%-1.9%
```

**Fig 11. Output from the function `frag.mas()`.**

```
> plot(out.mas, max.f = 40, cex.name = 0.5)
> plot(out.mas, fragility = "FQ", breaks = 20)
```

The first line produces the bar plot of FIs of all 564 MAs in Fig 12a, and the second line produces the histogram of FQs in Fig 12b. As displayed in the output of out.mas, the FIs may take large values up to 167, so max.f is specified as 40 for truncation. Most MAs have FIs less than 15 and FQs less than 1%.

## Discussion

This article has reviewed methods for assessing and visualizing the fragility of an individual study, pairwise MA, and NMA with a binary outcome; the package "fragility" is designed for implementing these methods. We have focused on introducing the usage of many user-friendly functions provided by this package and illustrating them via several worked examples.

The FI and FQ are useful tools to assess clinical results' fragility; many researchers are becoming interested in these measures due to the growing concerns about research reproducibility and replicability. Nevertheless, it may be limited to assess the fragility based entirely on the numerical value of the FI or FQ. Most existing software programs do not provide much additional information about the FI or FQ besides its numerical value. The package "fragility" offers a variety of results that may aid the assessment of fragility. For example, the package produces various plots that visualize the studies and treatments involved in event status modifications in the iterative process for computing the FI or FQ of a pairwise MA or NMA. Such plots may indicate studies or treatments that are potentially influential in the meta-analytic results; clinicians may carefully examine the reliability (e.g., methodological quality) of the associated studies. In addition, users can specify certain directions of event status modifications in each treatment group for an individual study. The package provides information about different scenarios when the significance or non-significance is altered. It is crucial to incorporate such detailed information with clinicians' opinions on a case-by-case basis; for some rare diseases, it may be more sensible to modify events to non-events.

There are still several limitations of the "fragility" package. First, using this package requires researchers to possess some coding skills; clinicians without coding training might not directly use this package for assessing the fragility of clinical studies. Second, the current version of the

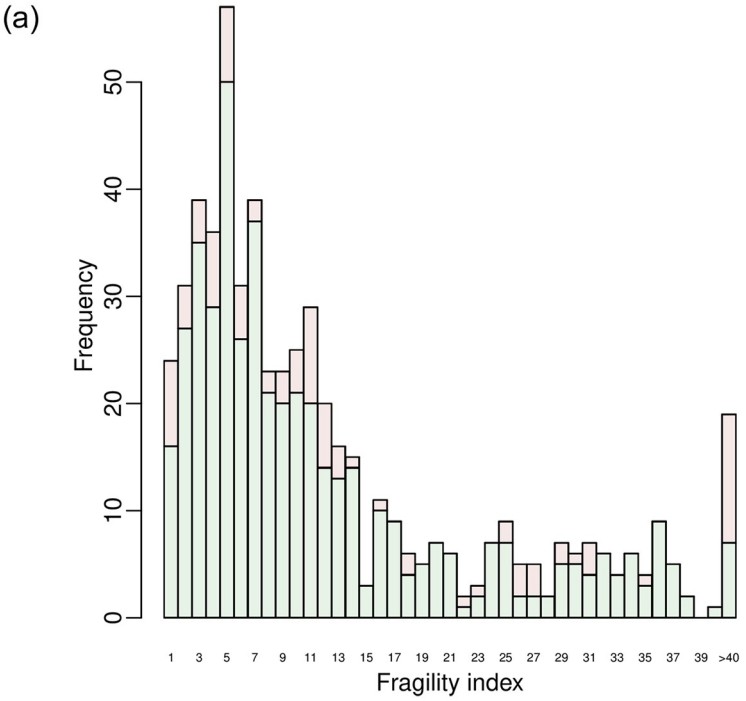

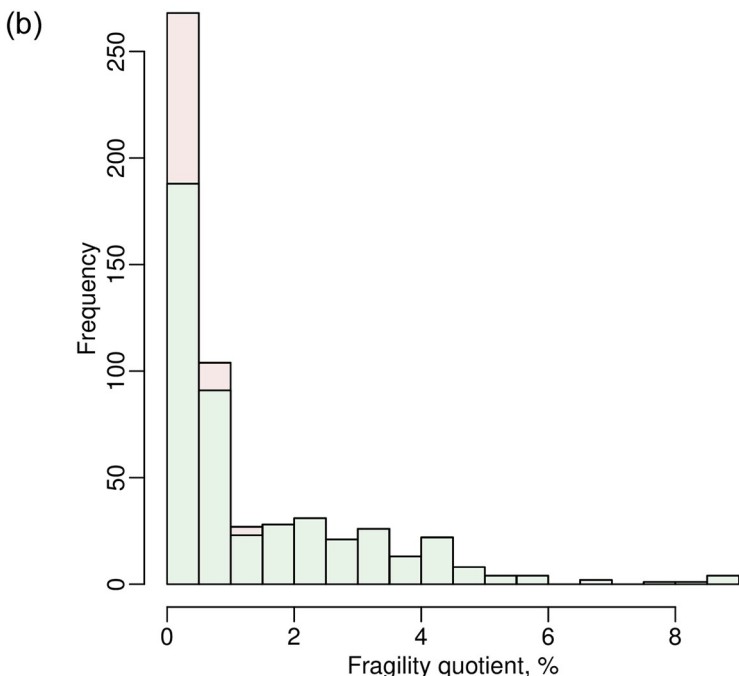

**Fig 12. Distributions of fragility measures of the pairwise meta-analyses in the dataset `dat.ns`.** (a) Bar plot of fragility indexes. (b) Histogram of fragility quotients.

"fragility" package can only deal with clinical studies with binary outcomes. Recently, Caldwell et al. [100] proposed a method to define the FI of studies with continuous outcomes. If this method is further validated in practical applications, we will consider incorporating it into the future releases of the "fragility" package. Third, the existing literature lacks a guideline or rule of thumb to interpret the magnitude of the FI or FQ (i.e., the extent of fragility). On the one hand, the interpretation might depend on the clinical setting, e.g., whether the outcomes of some patients are possibly modified. On the other hand, in future work, we plan to systematically collect many clinical studies, pairwise MAs, and NMAs across different specialties (e.g., from the Cochrane Library), obtain their FIs and FQs, and derive the empirical distributions for all datasets and those within subgroups of specific research areas. Such empirical distributions will be incorporated in future versions of "fragility," and they will further assist users in properly interpreting the fragility of clinical results.

## Supporting information

**S1 File. Additional information of functions' arguments.**
(PDF)

## Author Contributions

**Conceptualization:** Lifeng Lin.

**Formal analysis:** Lifeng Lin.

**Funding acquisition:** Haitao Chu.

**Methodology:** Lifeng Lin.

**Software:** Lifeng Lin.

**Visualization:** Lifeng Lin.

**Writing – original draft:** Lifeng Lin.

**Writing – review & editing:** Lifeng Lin, Haitao Chu.

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
