## [Decision Letter · Decision Letter 0]

23 Nov 2021

PONE-D-21-32479Assessing and visualizing fragility of clinical results with binary outcomes in R using the fragility packagePLOS ONE

Dear Dr. Lin,

Thank you for submitting your manuscript to PLOS ONE. After careful consideration, we feel that it has merit but does not fully meet PLOS ONE’s publication criteria as it currently stands. Therefore, we invite you to submit a revised version of the manuscript that addresses the points raised during the review process.

We look forward to receiving your revised manuscript.

Kind regards,

Paul Aurelian Gagniuc, PhD

Academic Editor

PLOS ONE

Journal Requirements:

“This research was supported in part by the U.S. National Institutes of Health/National Library of Medicine grant R01 LM012982 (LL and HC) and National Institutes of Health/National Center for Advancing Translational Sciences grant UL1 TR001427 (LL). The content is solely the responsibility of the authors and does not necessarily represent the official views of the National Institutes of Health.”

“This research was supported in part by the U.S. National Institutes of Health/National Library of Medicine grant R01 LM012982 (LL and HC) and National Institutes of Health/National Center for Advancing Translational Sciences grant UL1 TR001427 (LL). The content is solely the responsibility of the authors and does not necessarily represent the official views of the National Institutes of Health.”

“This research was supported in part by the U.S. National Institutes of Health/National Library of Medicine grant R01 LM012982 (LL and HC) and National Institutes of Health/National Center for Advancing Translational Sciences grant UL1 TR001427 (LL). The content is solely the responsibility of the authors and does not necessarily represent the official views of the National Institutes of Health.”

Reviewers' comments:

Reviewer's Responses to Questions

**Comments to the Author**

1. Is the manuscript technically sound, and do the data support the conclusions?

Reviewer #1: Yes

Reviewer #2: Yes

Reviewer #3: Yes

2. Has the statistical analysis been performed appropriately and rigorously? 

Reviewer #1: Yes

Reviewer #2: Yes

Reviewer #3: Yes

3. Have the authors made all data underlying the findings in their manuscript fully available?

Reviewer #1: Yes

Reviewer #2: Yes

Reviewer #3: Yes

4. Is the manuscript presented in an intelligible fashion and written in standard English?

Reviewer #1: Yes

Reviewer #2: Yes

Reviewer #3: Yes

5. Review Comments to the Author

Reviewer #1: The Fragility Index (FI) is an intuitive statistical metric used primarily in the medical literature to assess the statistical significance of study results. Lin et a. has developed an R package to calculate the frailty index in an user-friendly way. It allows both to calculate the FI of the meta-analysis (MA) to compare couple of treatments at a time, e.g. the efficacy of a drug versus placebo - and Network-Meta-Analysis (NMA), to compare multiple treatments - for example the efficacy of multiple drugs.

The first part of the article is devoted to the mathematical definition of the FI index and provides a brief overview of the methods commonly used to calculate the pvalue (Fisher's exact test, chi-squared, OR, RR and RD), and to quantify the FI for multiple levels of significance. The second part presents the functions of the R package and the applications to (i) individual clinical study (ii) pairwise meta analysis (iii) network meta analysis (iv) fragility at multiple levels of significance (v) fragility of multiple datasets .

The paper is overall well written, although the following points would deserve to be addressed in addition to the more minor points mentioned below.

Majors:

* section “Assessing Fragility” (from line 375 to line 723). Here the authors describe the functions of the R package with all the options. While this is interesting when using the package, it’s really hard to follow this section when reading the paper. Moreover It seems redundant with the documentation available at https://cran.r-project.org/web/packages/fragility/fragility.pdf. The authors should consider rewriting this section, focusing on a few applications rather than listing all possible options .

* Fig 2. is not clear. The authors should give more details on how the plot is generated and how to interpret it. Same comment for Fig.3

* section "Example of assessing fragility of mutliple dataset" (from line 1004 to line 1254). Here the authors plot the fragility measures of all trials. I find it difficult to appreciate the novelty of this section as the authors describe how to make the plots, but not how to interpret them. The authors should should consider rewriting this section.

Minors:

*There is no link to the CRAN page of the package (and so it is not easy to find the documentation) and it's not explained how to install it.

*Caption of Fig.1 is not clear. The authors should expand it and provide more details

Reviewer #2: This was an interesting manuscript describing a new R package for calculating and visualizing fragility of clinical studies, with clear worked examples to introduce the usage of the package.

I have a few questions/comments which, if addressed, would hopefully improve the manuscript.

1. Page 18: The author showed an example of an individual clinical study with demonstration of the number of modifications in each group for achieving non-significance altered from significance. However, in my perspective, the direction of modification in each group is less clear by indicating the number of non-events changed in group 0 and the number of events changed in group 1. For example, maybe it would be clearer to demonstrate the number of events add/minus in group 0 and the number of events add/minus in group 1?

2. Page 38: “Similar to assessing the fragility of an individual study and a pairwise MA, the FI is

defined as NA if the significance or non-significance cannot be altered.” Suppose we have three groups A,B and C, if the significance of comparing B vs A (or non-significance) cannot be altered by modifications in groups A and B, will you further consider modifications in group C or directly define it as NA. Maybe worth notify this somewhere.

3. Page 7: The explanation of deriving FI for MA under each step is not intuitive in my perspective. Could you please demonstrate a simple example here or illustrate a few iterations in the Example of a pairwise meta-analysis section?

4. The authors considered several tests with calculation of the corresponding FI. In practice, should we expect the FI to be similar across these tests? If the FI is very different among these tests, what does it imply?

Reviewer #3: The authors introduce an R package called fragility. The package contains functions and visualization tools that aid researchers in calculating the fragility index of their findings. An FI is a number that indicates how many observations must be shifted from one study sample to another for the statistical conclusion of the study to change. For example, if FI = 3, then three observations have to move from "event" to "non event" in order for a study in which the null hypothesis was rejected to become a study in which the null hypothesis is not rejected.

The package is necessary and the write up describes it in great detail - sometimes too much detail. The manuscript is 44 pages long (including references and supplements). A large proportion of the manuscript gives painstaking instructions on the functions in the fragility package and their many arguments and options for those arguments.

I request a reorganization of the paper. I would like to see Results section organized as a section on "Use Cases". By use cases, I mean having a case where the fragility index indicates an extremely fragile result (for all types of analyses - single study, MA, and NMA), and where the FI indicates a moderately fragile result, and where the FI indicates a robust result. The authors spend a great deal of white space in explaining the arguments of the functions. The reading is rather soporific. It would be better to show how the output and plots change as the FI changes for each of the three types of designs. That also means rearranging the plots. For example, a plot like Figure 1(a) for a fragile study should be placed next to the same rendering for a moderate and robust study. Such placement would help the reader understand what features of the plot are important.

I would like to see better explication of the plots (i.e. Figure 1). Perhaps this could be done with more detailed captions. I found it hard to examine the plots and the text, and then go back and forth to determine what features about the plots are being described in the text. I spent a few minutes with each one and I am still not sure that I really understand them. It seems to me that having clear, easy to explain, output for researchers is part of the reason this package exists. If the plots are hard to understand, then are they serving the researchers using this package?

A note on plotting symbols and colors: the triangles representing original p-values are quite small. I thought they were scratches on my screen at first. Finally, red and green is not a good color combination to show contrast. Red/green colorblindness is the most common type in the adult population. RColorBrewer has a list of color combinations that are meant to show divergence. One of those should be used.

I'm also a little concerned that the package, or the FI itself, seems to address only binary outcomes. The examples are about observations moving from events to non-events, but what if the measurements are quantitative and they change slightly? Such a situation might happen in the case of an outlier in the data. If the outlier were removed, how does that affect the fragility of the outcome? That said, the package is useful as it currently stands.

I think that researchers need to know about this package. It claims to make usage of FI easier, but as with all things R, there is a steep learning curve. A Shiny app would be much more helpful.

Minor Details:

The sentence at line 8 needs to be reworded. I think that "greatly" is the wrong adverb here, but I am not quite sure what the authors are trying to say; therefore, I can't suggest another word.

Line 1267 contains a sentence about user specification in the function call, but it follows a sentence about the many aids in assessment contained in the fragility package. I would change the sentence to one that describes the detailed output or the plots that are available.

I attached a PDF of the paper with minor grammatical and syntactical adjustments. They are not comprehensive. For example, the authors use the word "so" quite often as a conjunction. However, it isn't. "therefore" or "hence" or no conjunction at all should be used instead.

6. PLOS authors have the option to publish the peer review history of their article (what does this mean?). If published, this will include your full peer review and any attached files.

Reviewer #1: No

Reviewer #2: No

Reviewer #3: No

---

## [Author Response · Author response to Decision Letter 0]

13 Apr 2022

We thank the reviewers for providing many helpful comments on our manuscript. We have incorporated these suggestions in the revised manuscript. Our responses are detailed in the [square brackets] as follows.

Reviewer #1:

The Fragility Index (FI) is an intuitive statistical metric used primarily in the medical literature to assess the statistical significance of study results. Lin et a. has developed an R package to calculate the frailty index in an user-friendly way. It allows both to calculate the FI of the meta-analysis (MA) to compare couple of treatments at a time, e.g. the efficacy of a drug versus placebo - and Network-Meta-Analysis (NMA), to compare multiple treatments - for example the efficacy of multiple drugs.

The first part of the article is devoted to the mathematical definition of the FI index and provides a brief overview of the methods commonly used to calculate the pvalue (Fisher's exact test, chi-squared, OR, RR and RD), and to quantify the FI for multiple levels of significance. The second part presents the functions of the R package and the applications to (i) individual clinical study (ii) pairwise meta analysis (iii) network meta analysis (iv) fragility at multiple levels of significance (v) fragility of multiple datasets .

The paper is overall well written, although the following points would deserve to be addressed in addition to the more minor points mentioned below.

[Response: Thank you very much for taking the time to perform the review of our manuscript. We appreciate your positive comments regarding this manuscript and many helpful suggestions. We have incorporated your suggestions into the revised manuscript. Please see our detailed responses below.]

Majors:

* section “Assessing Fragility” (from line 375 to line 723). Here the authors describe the functions of the R package with all the options. While this is interesting when using the package, it’s really hard to follow this section when reading the paper. Moreover It seems redundant with the documentation available at https://cran.r-project.org/web/packages/fragility/fragility.pdf. The authors should consider rewriting this section, focusing on a few applications rather than listing all possible options.

[Response: Thanks a lot for this suggestion. In the revised manuscript, we have moved some details of less important arguments to the supplemental file “Additional information of functions' arguments.” We have reduced the main content devoted to describing the functions. Also, to group the descriptions of the functions more clearly, we have split the original subsection into three subsections for individual studies, pairwise meta-analyses, and network meta-analyses accordingly. Although the manual of our package also lists the meaning of each argument of each function, we try to explain them in more detail so that they are more accessible for users.]

* Fig 2. is not clear. The authors should give more details on how the plot is generated and how to interpret it. Same comment for Fig.3

[Response: Thanks for the suggestion. We have added the following details to the captions of Fig 2 and Fig 3 (now Fig 4 and Fig 6): “Each line represents changes in the total event count in a certain group. The numbers around each line indicate the studies involved in the event status modifications during the iterations; each asterisk indicates that a study remains unchanged as in the previous iteration.”]

* section "Example of assessing fragility of mutliple dataset" (from line 1004 to line 1254). Here the authors plot the fragility measures of all trials. I find it difficult to appreciate the novelty of this section as the authors describe how to make the plots, but not how to interpret them. The authors should should consider rewriting this section.

[Response: This article focuses on introducing how to use the “fragility” R package and implementing case studies. Considering the manuscript’s length, we did not plan to give too many interpretations, as these have been done in many earlier works on empirical analyses of FI, including the original paper by Walsh et al. that proposed the FI. In the revised manuscript, we have added some interpretations in the section “Example of assessing fragility of multiple dataset.” For example, we added: “The FIs range from 1 to 19; many trials have FIs between 1 and 10, implying that their significance could be altered by changing the event statuses of at most 10 patients.” Also, as mentioned in the manuscript, the current literature still has some debates about the FI’s interpretation, so we have tried to avoid being involved in such debates. We have included these as a limitation in the discussion section.]

Minors:

*There is no link to the CRAN page of the package (and so it is not easy to find the documentation) and it's not explained how to install it.

[Response: Thanks for this comment. The link was originally provided in the reference list. To make it easier for users to access this package and install it, we have added more details at the beginning of the subsection “Using the R package fragility.” In short, users can easily install the package in R using the command install.packages("fragility").]

*Caption of Fig.1 is not clear. The authors should expand it and provide more details

[Response: In the revised manuscript, we have added the following details to the caption of Fig 1 (now Fig 2). “Points or areas in green indicate non-significant results, and those in red indicate significant ones. Dashed lines represent no modifications in the corresponding groups. Square points represent the original p value, and triangle points indicate minimal modifications that alter the significance. In panels (a) and (d), event statuses are modified in both groups. Each point represents the extent of the p value corresponding to a specific combination of event status modifications in groups 0 and 1. In panels (c) and (d), event statuses are modified in only one group. Plus points represent truncated p values at 10^{−10}.”]

Reviewer #2:

This was an interesting manuscript describing a new R package for calculating and visualizing fragility of clinical studies, with clear worked examples to introduce the usage of the package.

I have a few questions/comments which, if addressed, would hopefully improve the manuscript.

[Response: Thanks a lot for taking the time to perform the review. Please see our responses as follows.]

1. Page 18: The author showed an example of an individual clinical study with demonstration of the number of modifications in each group for achieving non-significance altered from significance. However, in my perspective, the direction of modification in each group is less clear by indicating the number of non-events changed in group 0 and the number of events changed in group 1. For example, maybe it would be clearer to demonstrate the number of events add/minus in group 0 and the number of events add/minus in group 1?

[Response: Thanks for this comment. The idea of the fragility index is based on modifying event status in some treatment groups. The total samples remain unchanged; we only modified the event status (from events to non-events or from non-events to events) for some samples. Therefore, in our function’s output, we use languages like “inversing status of certain events or non-events in certain groups,” instead of “adding or reducing certain events.”]

2. Page 38: “Similar to assessing the fragility of an individual study and a pairwise MA, the FI is

defined as NA if the significance or non-significance cannot be altered.” Suppose we have three groups A,B and C, if the significance of comparing B vs A (or non-significance) cannot be altered by modifications in groups A and B, will you further consider modifications in group C or directly define it as NA. Maybe worth notify this somewhere.

[Response: Many thanks for raising this question. We have mentioned that we only consider event status modifications in the relevant two groups for a treatment comparison because of the long run time for multiple NMAs via iterations. In the revised manuscript, we have added cautionary notes following this sentence: “Of note, as mentioned above, the calculation of the NMA’s FI for comparison B vs. A is based only on modifying event status in groups A and B. It is possible that the change of significance cannot be achieved by any event status modification in groups~A and B, but it could be achieved by modifications in other groups. Therefore, users should interpret an FI value of NA in the contexts of the event status modifications in the relevant two groups A and B only.”]

3. Page 7: The explanation of deriving FI for MA under each step is not intuitive in my perspective. Could you please demonstrate a simple example here or illustrate a few iterations in the Example of a pairwise meta-analysis section?

[Response: Thanks for this comment. We agree that the only texts may not illustrate the iterative algorithm for deriving the FI of a meta-analysis very clearly. Initially, we planned to illustrate the process, but this manuscript has already contained many figures, so we eventually decided not to include additional visualizations for the iterative algorithm, and this manuscript emphasizes introducing the “fragility” package. We have also offered a worked example of a meta-analysis, and the full iterations can be seen in Fig 4 (previous Fig 2). In the revised manuscript, at the end of the relevant paragraph, we added the following sentences. “For visualizations of the iterative steps for deriving the FI of the MA, readers may refer to Fig 4 in Atal et al. We will also provide a worked example later to demonstrate the process.”]

4. The authors considered several tests with calculation of the corresponding FI. In practice, should we expect the FI to be similar across these tests? If the FI is very different among these tests, what does it imply?

[Response: We provide multiple methods for calculating p-values and thus deriving the FI to offer flexibility for users if they want to use different methods. However, we are not suggesting that users should try all methods and compare the resulting FI values. Researchers are commended for using the method specified in the original study’s protocol for deriving the FI. In the subsection “Fragility of an individual clinical study” of the revised manuscript, we have added the following paragraph. “Of note, this article discusses multiple methods for testing the association between treatment and outcome and thus deriving the FI. We hope that they offer flexibility for users when assessing clinical studies' fragility based on different tools. We do not suggest that users should try all methods because this practice could lead to “fragility-hacking.” Users are recommended to use the statistical method specified in the study protocol for deriving the FI.”]

Reviewer #3:

The authors introduce an R package called fragility. The package contains functions and visualization tools that aid researchers in calculating the fragility index of their findings. An FI is a number that indicates how many observations must be shifted from one study sample to another for the statistical conclusion of the study to change. For example, if FI = 3, then three observations have to move from "event" to "non event" in order for a study in which the null hypothesis was rejected to become a study in which the null hypothesis is not rejected.

The package is necessary and the write up describes it in great detail - sometimes too much detail. The manuscript is 44 pages long (including references and supplements). A large proportion of the manuscript gives painstaking instructions on the functions in the fragility package and their many arguments and options for those arguments.

[Response: Thank you very much for providing many helpful suggestions on our manuscript. We have substantially restructured the manuscript. Please see our detailed responses below.]

I request a reorganization of the paper. I would like to see Results section organized as a section on "Use Cases". By use cases, I mean having a case where the fragility index indicates an extremely fragile result (for all types of analyses - single study, MA, and NMA), and where the FI indicates a moderately fragile result, and where the FI indicates a robust result. The authors spend a great deal of white space in explaining the arguments of the functions. The reading is rather soporific. It would be better to show how the output and plots change as the FI changes for each of the three types of designs. That also means rearranging the plots. For example, a plot like Figure 1(a) for a fragile study should be placed next to the same rendering for a moderate and robust study. Such placement would help the reader understand what features of the plot are important.

[Response: Thanks a lot for this suggestion. In the revised manuscript, we have moved some details of less important arguments to the supplemental file “Additional information of functions' arguments.” We have reduced the main content devoted to describing the functions. Also, to group the descriptions of the functions more clearly, we have split the original subsection into three subsections for individual studies, pairwise meta-analyses, and network meta-analyses accordingly. Although the manual of our package also lists the meaning of each argument of each function, we try to explain them in more detail so that they are more accessible for users.

As for the suggestion of organizing the results section as “use cases,” we think it might be beyond this manuscript’s goal. This manuscript focuses on introducing how to use the “fragility” R package and implementing case studies. Considering the manuscript’s length, we did not plan to give too many interpretations, as these have been done in many earlier works on empirical analyses of FI, including the original paper by Walsh et al. that proposed the FI. More importantly, as mentioned in the manuscript, the current literature still has some debates about the FI’s interpretation. We have tried to avoid being involved in such debates. As discussed in the last section, there are no widely accepted rules for claiming very fragile, moderately fragile, slightly fragile, and no fragile studies/meta-analyses based on the FI or FQ values. In the future, we plan to systematically apply the FI and FQ to many studies and meta-analyses and thus derive evidence-based guidelines for interpreting the magnitudes of the fragility measures.]

I would like to see better explication of the plots (i.e. Figure 1). Perhaps this could be done with more detailed captions. I found it hard to examine the plots and the text, and then go back and forth to determine what features about the plots are being described in the text. I spent a few minutes with each one and I am still not sure that I really understand them. It seems to me that having clear, easy to explain, output for researchers is part of the reason this package exists. If the plots are hard to understand, then are they serving the researchers using this package?

[Response: Thanks a lot for this suggestion. In the revised manuscript, we have added the following details to the caption of Fig 1 (now Fig 2): “Points or areas in green indicate non-significant results, and those in red indicate significant ones. Dashed lines represent no modifications in the corresponding groups. Square points represent the original p value, and triangle points indicate minimal modifications that alter the significance. In panels (a) and (d), event statuses are modified in both groups. Each point represents the extent of the p value corresponding to a specific combination of event status modifications in groups 0 and 1. In panels (c) and (d), event statuses are modified in only one group. Plus points represent truncated p values at 10^{−10}.” We have added the following to the captions of Fig 2 and Fig 3 (now Fig 4 and Fig 6): “Each line represents changes in the total event count in a certain group. The numbers around each line indicate the studies involved in the event status modifications during the iterations; each asterisk indicates that a study remains unchanged as in the previous iteration.”]

A note on plotting symbols and colors: the triangles representing original p-values are quite small. I thought they were scratches on my screen at first. Finally, red and green is not a good color combination to show contrast. Red/green colorblindness is the most common type in the adult population. RColorBrewer has a list of color combinations that are meant to show divergence. One of those should be used.

[Response: As for the triangles in Fig 1a and Fig 1d (now Fig 2a and Fig 2d), we have slightly increased the point sizes. Please kindly note that the point sizes can be adjusted using the argument cex.pts in the function plot.frag.study.all().

As for the colors, users can change the colors for significance and non-significance using the argument col.sig in the function plot.frag.study.all(). We respectively argue that it is quite common to use red/green to show contrast in some scientific disciplines, at least in the literature of evidence synthesis. For example, in the assessment of the risk of bias, green is typically used for studies with a low risk of bias, and red is typically used for studies with a high risk of bias. See, e.g., https://doi.org/10.1136/bmj.l4898. In PLOS ONE, many articles also use red/green for contrasts; see, e.g., https://doi.org/10.1371/journal.pone.0217401.g003 and https://doi.org/10.1371/journal.pone.0025389.g002.]

I'm also a little concerned that the package, or the FI itself, seems to address only binary outcomes. The examples are about observations moving from events to non-events, but what if the measurements are quantitative and they change slightly? Such a situation might happen in the case of an outlier in the data. If the outlier were removed, how does that affect the fragility of the outcome? That said, the package is useful as it currently stands.

[Response: Thanks for this comment. In the current practice of using the FI, nearly all studies deal with binary outcomes. Recently, Caldwell et al. (https://doi.org/10.1016/j.jclinepi.2021.02.023) proposed a method to calculate the FI for continuous outcomes. Nevertheless, to our knowledge, this method has not been widely used for assessing the fragility of studies with continuous outcomes. In the revised manuscript, we have added the following as one of our limitations in the discussion section: “Second, the current version of the “fragility” package can only deal with clinical studies with binary outcomes. Recently, Caldwell et al. proposed a method to define the FI of studies with continuous outcomes. If this method is further validated in practical applications, we will consider incorporating it into the future releases of the “fragility” package.”]

I think that researchers need to know about this package. It claims to make usage of FI easier, but as with all things R, there is a steep learning curve. A Shiny app would be much more helpful.

[Response: We agree that using this R package requires researchers to possess some coding skills. As mentioned in the introduction, some webpages (e.g., https://clincalc.com/Stats/FragilityIndex.aspx and https://clinicalepidemio.fr/fragility_ma/) can be used to derive the FI for an individual study and a meta-analysis. Nevertheless, our R package “fragility” offers many more additional features than these webpage-based tools. If users just want to obtain numeric values of the FI, they could use the webpage-based tools. If they would like to make further analyses and better learn the fragility of clinical studies, they could consider using our package. Developing a Shiny app for implementing all the features offered by our current “fragility” R package is currently beyond our expertise. To the best of our knowledge, no Shiny app is available for implementing network meta-analyses, so it’s unlikely that a Shiny app for deriving the FI for NMA could be developed in the near future. In the last paragraph of the discussion section, we have added the following as our limitations. “There are still several limitations of the ``fragility'' package. First, using this package requires researchers to possess some coding skills; clinicians without coding training might not directly use this package for assessing the fragility of clinical studies.”]

Minor Details:

The sentence at line 8 needs to be reworded. I think that "greatly" is the wrong adverb here, but I am not quite sure what the authors are trying to say; therefore, I can't suggest another word.

[Response: Sorry that there was a typo in this sentence. We have changed it to “This bias may distort clinical conclusions toward an artificially favorable direction and thus greatly threaten their reliability.”]

Line 1267 contains a sentence about user specification in the function call, but it follows a sentence about the many aids in assessment contained in the fragility package. I would change the sentence to one that describes the detailed output or the plots that are available.

[Response: Thanks for this suggestion. We have changed the relevant sentences as follows. “For example, the package produces various plots that visualize the studies and treatments involved in event status modifications in the iterative process for computing the FI or FQ of a pairwise MA or NMA. Such plots may indicate studies or treatments that are potentially influential in the meta-analytic results; clinicians may carefully examine the reliability (e.g., methodological quality) of the associated studies.”]

I attached a PDF of the paper with minor grammatical and syntactical adjustments. They are not comprehensive. For example, the authors use the word "so" quite often as a conjunction. However, it isn't. "therefore" or "hence" or no conjunction at all should be used instead.

[Response: Thanks a lot for carefully reviewing the whole manuscript. We appreciate your corrections. We have made changes accordingly and double-checked the manuscript in the revision.]

---

## [Decision Letter · Decision Letter 1]

9 May 2022

Assessing and visualizing fragility of clinical results with binary outcomes in R using the fragility package

PONE-D-21-32479R1

Dear Dr. Lin,

We’re pleased to inform you that your manuscript has been judged scientifically suitable for publication and will be formally accepted for publication once it meets all outstanding technical requirements.

Kind regards,

Paul Aurelian Gagniuc, PhD

Academic Editor

PLOS ONE

Additional Editor Comments (optional):

Reviewers' comments:

Reviewer's Responses to Questions

**Comments to the Author**

1. If the authors have adequately addressed your comments raised in a previous round of review and you feel that this manuscript is now acceptable for publication, you may indicate that here to bypass the “Comments to the Author” section, enter your conflict of interest statement in the “Confidential to Editor” section, and submit your "Accept" recommendation.

Reviewer #1: All comments have been addressed

Reviewer #2: All comments have been addressed

2. Is the manuscript technically sound, and do the data support the conclusions?

Reviewer #1: Yes

Reviewer #2: Yes

3. Has the statistical analysis been performed appropriately and rigorously? 

Reviewer #1: Yes

Reviewer #2: Yes

4. Have the authors made all data underlying the findings in their manuscript fully available?

Reviewer #1: Yes

Reviewer #2: Yes

5. Is the manuscript presented in an intelligible fashion and written in standard English?

Reviewer #1: Yes

Reviewer #2: Yes

6. Review Comments to the Author

Reviewer #1: I would like to thank the authors for attempting to address my main concerns, and I think the revised paper is substantially improved.

I appreciated the effort they have done to improve the readability of paper by moving most of the technical details to the supplementary and leaving in the main text only the main functions.

They have added the captions and more extensive explanations to Fig.2 and 3 and some interpretations of the Fragility Index which in my opinion are of great use to non-specialists.

Also all the minor comments have been addressed.

Reviewer #2: The paper has been written in a more concise way. I am satisfied with the work done by the authors to address my questions.

7. PLOS authors have the option to publish the peer review history of their article (what does this mean?). If published, this will include your full peer review and any attached files.

Reviewer #1: No

Reviewer #2: No

---

## [Editor Report · Acceptance letter]

11 May 2022

PONE-D-21-32479R1 

Assessing and visualizing fragility of clinical results with binary outcomes in R using the fragility package 

Dear Dr. Lin:

I'm pleased to inform you that your manuscript has been deemed suitable for publication in PLOS ONE. Congratulations! Your manuscript is now with our production department. 

Kind regards, 

on behalf of

Dr. Paul Aurelian Gagniuc 

Academic Editor

PLOS ONE